# Environmental Management Framework for Road Network Demolition Wastes for Construction Industry of Pakistan

Sajjad Shuker Ullah [ID], Ishtiaq Hassan [ID] and Syed Shujaa Safdar Gardezi *[ID]

Department of Civil Engineering, Capital University of Science and Technology (CUST), Expressway, Kahuta Road Zone-V Sihala, Islamabad 44000, Pakistan; dce183001@cust.pk (S.S.U.); eishtiaq@cust.edu.pk (I.H.)
* Correspondence: dr.shujaasafdar@cust.edu.pk

**Abstract:** Demolition waste from construction industry, especially from road networks, is one of the most voluminous and harmful categories of waste worldwide; therefore, its proper handling is essential for sustainable waste management for environmental, social, and economic benefits. Prolific and unregulated construction activities, conflicts, and defective works are major reasons. The current work aims to address the issue by presenting a framework for an enhanced understanding of sustainable demolition waste management (DWM). A critical analysis of the literature aided to identify major concerns related to different causes, their impacts, and challenges being faced by the construction industry in such management endeavors. The study adopted questionnaire-based methodology to understand the critical relation among the three variables. The Delphi technique supported by industry professionals and pilot study helped to formulate a realistic questionnaire tool. Using the concept of multivariate statistical analysis, structure equation modeling (SEM) helped to assess the structural relationships between the three variables. The research instrument met the reliability, validity and internal consistency criteria required. Each variable achieved a high effect size, $f^2$, with a value of co-efficient of determination of more than the threshold value of 70%. Thus, this supported the fitness criterion of the SEM-based measurement model. Path coefficients yielded the acceptance of all alternate hypotheses, resulting in a strong positive relationship among the three constructs. Therefore, demolition waste impacts are deemed as an effective mediator when explaining the impact between the other two variables. The developed framework presents a coherent and systematic approach and identifies strategies that could be used to address these issues and lead to DWM, including options available for capacity building and implementation and evaluation for supporting sustainability.

**Keywords:** environmental management; framework; demolition waste; SEM

## 1. Introduction

Rapid population growth and subsequent urbanization contribute significantly to the increasing natural resource consumption. It promotes the development of roads, buildings, and demolition wastes (DWs), as well as the environmental impacts associated with it. The construction industry has made a major contribution in recent years to the rise in the illegal dumping of solid waste, particularly in developing countries, resulting in considerable environmental harm. The construction sector consumes a major chunk of renewable materials, and it generates massive amounts of waste in the waste stream. During or after the demolition work, most of the waste materials contain hazardous materials. The waste is made of bitumen, brass, cords, asbestos, light bulbs and fittings, wood preservatives, and heavy metal-containing concrete materials. These waste products also contain heavy metals. Hazardous contaminants, such as water, soil, and air, are emitted on waste disposal into different environmental platforms [1].

Demolition waste was classified into 38 subcategories under the European Waste Catalog (EWC). Of these subcategories, sixteen have been graded as dangerous in absolute

tests. Furthermore, in demolition waste samples, Zn concentration was the highest among heavy metals. The hazardous character of these waste materials depends heavily on the source of the area of formation [2]. The most significant waste material containing harmful substances, followed by bitumen, glass, and steel residue, was made up of cement, sand, and aggregates.

Demolition waste from roads represents a significant global sustainability challenge. Evidence suggests that approximately 40% of the total volume of waste generated worldwide originates from demolition activities [3]. In the European Union (EU), demolition waste accounts for approximately 25–30% of all types of waste generated [4]. A variety of operational definitions of waste are available for industry and academic use. In accordance with the new production philosophy, waste is defined as any inefficiency leading to the use of more equipment, material, labor, or capital than necessary in the production of a building, encompassing its entire lifespan, and often including demolition. Waste includes both material losses and unnecessary work that generates additional costs but does not contribute value to the output [5].

The construction sector is a highly wasteful economic sector with severe environmental implications; hence, there have been considerable initiatives to eliminate construction. The legislative body [6] defined construction waste as demolition waste substances resulting from various construction activities such as digging, development, demolition, damage, renovation, road work, and frequently a mixture of inert and non-inert materials.

The growing challenge of demolition waste is one of the pertinent problems that has intrigued researchers. The EU contributes 800 million tons [7] and about 2300 million tons come from China [8]. Demolition waste generated at the global level is more than 10 billion tons. Furthermore, both the United States and China are major economies that face problems in the handling of demolition waste. Due to exponential development in the building industry and urbanization, the United States contributes around 30 percent of the world's annual total demolition waste, while China accounts for approximately 30–40 percent [9].

Demolition waste can originate from various sources and causes during the demolition process. Some may result from design flaws that existed prior to project execution, while others could arise from ongoing modifications or market trends affecting the supply chain. To address the growing issue of demolition waste, numerous construction waste management strategies have been implemented to minimize waste generation and increase recycling rates [10]. It is now imperative to adopt principles of sustainable demolition waste management (DWM) to mitigate the potential adverse impacts associated with such waste in terms of economic, environmental, and public health dimensions [3].

In the construction industry, while road engineering does contribute to waste generation, it is not typically the primary source. Construction waste and tailings from various construction activities, such as building construction, demolition, and excavation, often form a significant portion of the waste generated.

However, there has been a growing trend in utilizing construction waste and tailings in subgrade engineering. This involves recycling and reusing these materials in the construction of the foundation layers of roads and other infrastructure projects. This practice not only helps in managing waste but also reduces the demand for natural resources and lowers overall project costs. Additionally, it can improve the mechanical properties and stability of the subgrade, leading to more sustainable and durable infrastructure [11]. Table 1 summarizes the key differences between construction waste management and road engineering waste management.

This research component explores the sources, impacts, and challenges associated with demolition waste, which lacks a distinct or fixed definition. Demolition waste can originate from various sources and reasons during the demolition process. Design faults existing before project execution, ongoing modifications, and market trends affecting the supply chain can all contribute to the generation of demolition waste [12].

**Table 1.** Construction waste management vs. road engineering waste management.

| Aspect | Construction Waste Management | Road Engineering Waste Management |
|---|---|---|
| Scope of activities | Encompasses waste from building construction, renovation, and demolition projects | Focuses on waste from road construction, maintenance, and demolition activities |
| Types of waste | Includes a wide range of materials such as concrete, wood, steel, plastics, and packaging | Primarily consists of materials like asphalt, aggregates, bitumen, and road markings |
| Regulatory considerations | Regulations vary based on environmental policies, waste management laws, and building codes | Adherence to road construction specifications, quality control measures, and environmental impact assessments |
| Technological solutions | Waste sorting systems, material recovery facilities, and advanced recycling technologies | Asphalt recycling plants, mobile crushing units, and specialized pavement rehabilitation technologies |
| Environmental impacts | Resource depletion, habitat destruction, landfill space consumption, and pollution | Air and water pollution, habitat fragmentation, and soil erosion |
| Goal | Minimize waste and environmental impacts through waste reduction, reuse, and recycling | Mitigate environmental impacts while promoting resource efficiency and pollution prevention |

Waste can generally be broken down to non-physical and physical wastes. Physical waste is generated by the different construction activities of concrete, aggregates, asphalt, sand, timbers, metals, and plastics. Non-physical waste refers to waste that is not tangible or material in nature. Instead, it typically involves aspects such as time, energy, or resources that are lost or mismanaged in various processes. The production of demolition wastes is caused by several factors including inaccurate handling, piling, cutting, and retention of construction equipment, insufficient attention given to the measurement of components, inadequate building knowledge during planning, and lack of confidence among contractors. Routinely, approximately 1–10% of supplies are dumped as waste at the premises depending on the material category.

Materials may have a negative effect on human beings and the environment during execution, alteration, and demolition activities [13,14]. These effects are high priority in terms of land degradation, air pollution, and polluted and poisonous buildings, and there is an immediate need to minimize wastage and to reduce the environmental impact [15]. The recycling of demolition waste (DW) is considered to be the preferred strategy in the United States and a few countries of Europe (EU) where concrete resources are scarce, and several concrete waste recycling and sorting plants have been established [16]. The negative impacts of DW waste on the environment have become a worldwide problem [17]. At the same time, DW is associated with a series of environmental implications. Therefore, it is urgently necessary to minimize the environmental impact of DW.

Efficient data collection is essential for effective demolition waste management in road projects. Gathering comprehensive information about the causes of demolition waste supports research across the project life cycle (PLC), including planning, execution, and monitoring and control stages [18]. According to the Waste and Resources Action Programme (WRAP), stakeholders such as the owner or client, consultants, and contractors play crucial roles in either reducing or increasing the effectiveness of demolition waste management [19]. The responsibilities of stakeholders in completing a project are as follows: the owner or client provides the necessary funds and conceptual idea for the project, consultants are tasked with estimating, designing, and overseeing the project, and the contractor carries out the construction work based on the provided drawings, designs, and specifications [20,21]. The stages of the product life cycle (PLC) are essential because they provide insight into the factors contributing to demolition waste generation, thereby assisting stakeholders in understanding and addressing this issue in their daily practices. Moreover, initially, shortcomings in the bidding process or in the bidding documents can lead to waste generation during the planning stage [22,23]. Estimating the demolition quantity accurately at the initial stage can significantly improve project efficiency and outcomes [24,25]. The absence of a demolition waste management process in the bidding

documents for a road project does not legally compel the execution parties to implement such a process. The absence of a demolition waste management process in the bidding documents for a road project can indeed create challenges in effectively managing demolition waste during project execution. Without clear guidelines or requirements in the bidding documents, contractors and project managers may face difficulties in implementing proper waste handling practices. This could result in inefficient waste management, potential environmental impacts, increased project costs, and compliance issues with waste disposal regulations. Therefore, including detailed waste management specifications in the bidding documents is crucial for ensuring that demolition waste is handled responsibly and effectively throughout the project life cycle [26]. Furthermore, incomplete estimates, drawings, and bidding documents can contribute to waste generation during a project. When these essential documents are not comprehensive or accurate, it can lead to misunderstandings, rework, and inefficiencies during construction or demolition activities. Contractors may encounter unexpected challenges or discrepancies, which can result in material waste due to incorrect or excessive ordering, revisions, or adjustments to accommodate the missing information. Therefore, ensuring that estimates, drawings, and bidding documents are thorough and detailed can help minimize waste and improve the overall project efficiency [27].The design of a project is critical for achieving optimal outcomes by involving all project stakeholders during the planning stage of a road project. This involvement helps concentrate efforts on selecting appropriate sustainable materials [28,29]. Many researchers emphasize that the success of a project depends significantly on its design and execution. If there are changes observed in the design or scope of the project, the project may suffer. Similarly, changes in these aspects can also impact the waste produced during the project [30,31]. A significant drawback in project administration is often the lack of proper documentation throughout the project life cycle, from the initial steps to project delivery. This guidance should be emphasized for upcoming projects to ensure thorough and comprehensive documentation at every stage of the project [32,33].

According to many researchers, demolition waste majorly depends upon the gap between the stakeholders to overrun the time, cost, and scope [23,30]. Secondly, waste generation is also influenced by frequent changes in project scope or last-minute alterations, which can contribute to inefficiencies and increased waste during project execution [27]. Frequent changes in project scope can disrupt or even lead to neglect of the waste management plan, which can significantly contribute to increased waste generation [27,29]. A large amount of waste is generated due to a lack of properly trained staff and labor, as well as the inadequate provision of necessary equipment. Strict supervision can help reduce waste generation according to research sources [26,31]. Additionally, proper storage, salvaging, or returning of waste or remaining materials can also contribute to reducing waste generation, based on findings from research sources [25,34].

The cause of demolition waste primarily hinges on the timely identification and quantification of waste, as noted by research [27,34]. This waste can be reduced by leveraging Information Technology (IT) for waste identification and quantification, as supported by studies [32,35,36]. Subsequently, sorting the waste based on type using suitable methods or techniques aligned with site capabilities can further aid in waste reduction, as highlighted by research [34,37]. After sorting the waste, the next critical step involves the proper storage and handling of the waste, as discussed in studies [30,34,38]. Another significant factor contributing to waste generation is the practice of subcontract awarding or sub-letting, which can result in quality compromises and the need for redoing work due to the use of substandard materials, ultimately leading to waste and time consumption [28,38]. Implementing the principles of 3R (reduce, reuse, recycle) is an effective strategy for waste minimization, as recommended by many researchers in their study inferences [39,40]. Table 2 shows the details of various variables that cause demolition waste.

**Table 2.** Causes of demolition waste.

| Variable | References |
| --- | --- |
| Pre-demolition audit at planning stage | [24,25,29,41] |
| Role of supervision skills | [23,26,27,30,31,42] |
| Errors and omissions in contract documents and need to revise contractual clauses | [23,30,41] |
| Promoting sustainable material while designing | [28,43] |
| Implementation of lessons learnt | [32,33] |
| Role of 3R (reduce, reuse, recycle) strategy | [39,40] |
| Waste generation due to poor workmanship | [27,30,31,44] |
| Discrepancies in bidding document | [23,30,41] |
| Incorporation of demolition waste management in bidding process | [26] |
| Incomplete bidding documents before tendering | [27] |
| Scope/design changes | [27,30,31,41] |
| Identification and quantification of demolition waste | [27,30,34,38] |
| Coordination and communication among stakeholders | [27,29] |
| Utilization of substandard materials that result in wastage | [28,43] |
| Inadequacy of implementation in waste management plan | [23,25,27] |
| Inappropriate storage of unused construction materials | [27,30,34,45] |
| Impact of eleventh-hour change of scope | [27] |
| Coordination and communication gap among stakeholders | [23,26,30] |
| Role of IT in demolition waste management mechanism | [32,36] |
| Consideration of site storage and space availability | [25,30,34,38] |
| Impact of on-Site sorting techniques | [34,46] |
| Impact of sub-letting/subcontracting | [27,30,34] |
| Contract modification due to discrepancies | [27,47] |

Roads play a critical role in transportation networks, especially as more consumers rely on automobiles for daily travel. However, the extensive road network has significant detrimental environmental effects, including global warming, increased energy use, land transformation, and acidification [48]. The disposal of demolition waste in landfills and the consumption of resources such as green land lead to soil contamination [10,49] and water pollution [48,50] and also create dust generation and air contamination [47,51]. Secondly, demolition waste can leach harmful chemicals and minerals into groundwater, polluting underground water sources [46,49]. Thirdly, waste generation not only involves leftover building materials but also contributes to the wastage of natural resources, increasing energy consumption and the use of construction materials [24,52]. Fourthly, improper waste disposal and deposition instead of utilization contribute to energy consumption and greenhouse gas emissions through transportation [17,53]. Greenhouse gas emissions, particularly methane, indirectly contribute to global warming [28], adversely affecting animals and marine life [54]. Finally, the deposition of demolition waste on green land obstructs precipitation, which is crucial for replenishing the water table [39,46].

The social impact of demolition waste management is a key aspect of sustainability, which raises concerns about human health risks for residents living near dumping sites or areas where demolition waste is illegally disposed [29,43]. Moreover, ensuring the health and safety of individuals is paramount, and this can be compromised by the effects of demolition waste [55,56]. Sustainable development relies on addressing the social, environmental, and economic impacts caused by demolition waste [28,43]. Effective coordination and communication among stakeholders are vital for the success of any project, particularly in demolition waste management, where stakeholder attitudes are crucial [26,57]. Managing demolition waste is complex and requires additional human resources to maintain social viability [28,50,58], which also leads to increased energy consumption [57,58]. Demolition waste generation contributes to noise pollution and vibrations, but these impacts can be mitigated through proper mechanisms and by adopting specifications to minimize negative effects in nearby areas. These challenges often arise due to a lack of awareness regarding the social impacts of demolition waste management [36,59]. Raising awareness and motivating the general public and stakeholders to implement waste sorting, recycling, and

reuse practices can help reduce demolition waste and conserve natural resources [34,50]. Additionally, incorporating recycling or the reuse of materials into design considerations can preserve cultural aspects while minimizing waste generation [20].

Promoting recycled material and its utilization is the basic requirement, and its use can be accelerated by creating incentive for the utilization of recycled material, which is the first step towards a proper waste management practice [43]. Economic impact is associated with the cost, cost associated with the disposal of waste [38], cost associated with the management and operation [58], financial impact associated with creating awareness [28,43], and financial impact of recycling plants and stockpiled materials [28,58]. Promoting awareness and disseminating knowledge on the economic impact of waste by recycling or reuse [28,60] and providing designated landfill site for construction waste [56] are paramount and result in job creation [36,59]. Table 3 shows the detailed impacts of demolition waste reported by various researchers.

**Table 3.** Impacts of demolition waste.

| Variables | References |
|---|---|
| Barrier to recharge for water table | [39,43,46,54,60,61] |
| Air contamination due to pollution and dust generation. | [17,47,51,62] |
| Indirect impact on creation of climate change (e.g., methane gas, etc.) | [17,28,33,53,59,63,64] |
| Requirement of energy consumption | [17,28,50,57,58] |
| Vibrations and noise pollution impacting society | [28,60] |
| Financial impact of recycling plants and stockpiled materials | [28,58] |
| Impact of management and operation costs | [17,28,43,57,58] |
| Limited knowledge of waste recycling/reuse | [28,60] |
| Level of motivation for waste sorting/reusing | [34,50] |
| Consideration of conservative cultural aspects during construction design | [20] |
| Green house gas emissions | [17,28,47,53,59,63,64] |
| Leaching effect (extraction of soluble chemical and mineral carriers into liquid by rainwater) e.g., acid | [31,36,43,49,59,62,65] |
| Water contamination | [17,28,50,57,58,60,61] |
| Job creation opportunities | [36,43,59,62,65] |
| Natural resources consumption (i.e., construction material) | [24,43,52,66–69] |
| Green land utilization and soil contamination | [17,43,46,47,50,52,70] |
| Sustainable development | [28,43] |
| Impact of biodiversity (e.g., harm towards animal and marine life) | [17,28,47,53,59,63,64] |
| Additional human resource consumption | [28,50,58] |
| Health hazards in nearby communities | [29,34,43,55,56] |
| Health and safety impacts | [29,34,43,55,56] |
| Lack of awareness of social impacts of demolition wastes | [36,43,59,62,65] |
| Project stakeholders attitude towards DW management | [26,33,34,38,57] |
| Costs associated with disposal of waste | [26,33,34,38,57] |
| Resources required for creating designated dumping zone | [29,34,43,55,56] |
| Economic impact of additional incentive required for proper waste management | [29,34,43,55,56] |
| Financial aspect of creating awareness towards demolition management | [28,43] |
| Promotion and utilization of recycled materials | [71] |

The disposal of demolished materials poses significant challenges and negative effects on human beings. These challenges are addressed with a high priority through waste management strategies, awareness campaigns, waste handling training, and the implementation of policies and legislation. One of the primary issues contributing to demolition waste challenges is the illegal dumping of waste in nearby ditches or depressions [65,71]. A strict supervision and the allocation of a suitable landfill site is required before the start of the execution of the work [50]. Government agencies should focus and impose law and order, and strict actions may be initiated against the violators and also promote the awareness regarding environmental protection regulation to control and support government legal enforcement in implementation [27,72]. It is not only the duty of government agencies but also of construction companies to support and promote the effective utilization of recycled materials [26,33,43]. Effective waste management depends upon budget allocation

and helps to adopt strategic plans for effective demolition waste management [68,73]. Strategic plan preparation and implementation are purely dependent upon the close coordination and communication among stakeholders [67,73]; also, the acceptance of demolition waste as a recycling material by stakeholders instead of using new material is of equal importance [62,72]. For the effective management of demolition waste, it is important to identify specialized companies for waste handling and assign them specific tasks. This approach helps streamline the supply chain and reduce associated issues [74,75].

Waste management practices as outlined in contract documents are often overlooked, leading to waste being improperly disposed in nearby jungles, ditches, or depressions, which violates contractual agreements [23,26,27]. Training and courses in demolition waste management are essential to raise awareness and reduce the environmental impact of waste disposal [57]. Companies should prioritize in-house training opportunities for their officers and officials to maximize the effectiveness of demolition waste management efforts [42,57]. Training initiatives can significantly improve the performance of execution staff in waste management compared to traditional practices [26,33]. While changing existing practices and procedures may be challenging, experimenting with different methods and adopting the most successful approach can help overcome waste management challenges [25]. Addressing challenges in demolition waste management requires industry norms to be revised to prioritize cost and time savings alongside waste management efforts [74,76]. Additionally, improving supplier knowledge and cooperation towards using recycled materials helps conserve natural resources and mitigate environmental impacts associated with waste disposal [74], aligning with the role of technological support for smart construction [39].

Promoting the utilization of recycled or waste products and materials, along with providing incentives to encourage an industrial culture, benefits society [32,54]. Increasing the use of recycled materials can be achieved by leveraging all forms of social media to promote their equivalence with new or raw materials, ultimately supporting an environmentally friendly process [77]. Achieving the best results in utilizing recycled materials throughout the project life cycle can be accomplished by enforcing rules, regulations, or penalization mechanisms for generating demolition waste and damaging the environment at any stage of the project [78,79]. The role of legislation in enforcing policies to protect the environment has been highlighted in the literature [72]. Compliance with waste management plans as per contract or bidding documents is a legal requirement for contractors to execute in letter and spirit [66,79,80]. The enforcement of environmental protection laws and policies plays a crucial role in addressing waste challenges and promoting sustainability [26,42]. Furthermore, Table 4 summarizes various challenges to the appropriate solution for demolition waste management.

**Table 4.** Challenges of demolition waste.

| Variables | References |
| --- | --- |
| Arrangement of Designated landfill sites | [26,43,50,56] |
| Issues with supply chain management | [38,74,76] |
| Limited Specialized demolition waste handling companies | [33,50,69] |
| Knowledge of supplier's co-operation towards waste material utilization | [74] |
| Industry support for effective utilization of demolition waste | [26,33,43] |
| Role of awareness and training for environmental protection | [26,27,29,57] |
| Illegal dumping of demolition waste | [24,34,36,43,55,65,71,73] |
| Engaging all types of social media for promoting demolition waste management | [34,77] |
| Strategic plans for effective demolition waste management | [25,73] |
| Role of regulatory control and government legal enforcement | [27,75,76] |
| Support of existing practice and policies | [25] |
| Demolition waste management budget allocation | [26,33,68] |
| Acceptance of demolition waste as recycling material by stakeholders | [24,33,55,62,67,72] |
| Promotion of collaboration among stakeholders | [25,38,67,68,73] |

**Table 4.** *Cont.*

| Variables | References |
|---|---|
| Arrangement of in-house training on environmental management | [42,57] |
| Role of project executing staff towards demolition waste management | [26,33] |
| Role of legislation for enforcing policies to protect environment | [27,72] |
| Penalization mechanism for generating demolition waste and damaging environment | [26,43,60,78,81] |
| Demolition waste management consideration in project life cycle | [55,76,77,79] |
| Level of demolition waste management by trained staff and expert personnel as per contract document | [23,26,27] |
| Level of support by government agencies for environmental protection promotion | [72] |
| Enforcement of legal requirements on environmental protection | [26,34,38,42] |
| Industry culture of incentive policies to promote utilization of demolition waste management | [32,54,73] |
| Inadequacy of industry norms | [33] |
| Role of technological support for smart construction | [39] |
| Level of enforcement of waste management plan | [66,79,80] |
| Impact of industrial focus on cost and time rather than demolition waste management | [38,54,74,76] |

Sustainable construction is an important development strategy that takes into account the environment, society, economy, and culture. It is becoming increasingly important for green construction to study the link between the causes, consequences, and challenges of demolition waste in long-term sustainability.

Researchers have outlined several compelling reasons for using Partial Least Squares Structural Equation Modeling (PLS-SEM), particularly when faced with challenges such as limited sample sizes and model complexity in predictive modeling [82]. PLS-SEM is especially advantageous for handling complex models involving multiple constructs and relationships, allowing researchers to estimate and validate predictions effectively within the context of model complexity and limited sample size constraints.

One of the key strengths of PLS-SEM is its capability to perform well with small sample sizes, which is often encountered in business and marketing studies. It also demonstrates strong predictive capabilities, making it suitable for studies that require addressing model complexity while ensuring prediction accuracy. Furthermore, researchers are increasingly adopting PLS-SEM due to its ability to estimate and validate theoretically grounded models using constructs, its suitability for complex modeling scenarios with limited data, and its robust predictive performance [83–85]. PLS-SEM is viewed as a valuable analytical tool for exploring and understanding constructs and variables in research, presenting new avenues and opportunities for scholars across various fields.

*Originalities of the Research*

Based on the critical literature review, there exists a research gap for environmental management specific to road demolition waste. This becomes more relevant in the case of developing countries having rapid urbanization and population growth where road network is key to the social and the economic upliftment. The comprehensive management of all phases of the life cycle seems to be missing. A numerical correlation between various variables responsible for environmental management for infrastructure facilities, specifically road network, to support the sustainable demolition waste management practices is also missing.

Thus, the current work aims to support a robust framework by encompassing critical analyses, innovative methodologies, cross-disciplinary insights, and empirical validations, to promote a sustainable change for serious issues of waste management in the road construction industry.

## 2. Research Methodology

This research aimed to develop a framework for the demolition of roads by involving very experienced personnel. The primary data collection was validated and further analyzed.

The study used a questionnaire as a tool to measure its variables. The validation of this survey instrument was conducted in two stages. In the first phase, the face and content validity of the survey were evaluated by experts in the field. The second phase involved a pilot study to assess the instrument's reliability before the main study to test the survey's effectiveness and reliability in measuring the variables of interest [71,86].

The selection criteria for variables were based on publication years (2014–2024). Initially, 94 variables were identified from the literature. Based on the results of the frequency analysis, 87 variables were selected for further investigation and discussion in a focus group discussion (FGD) using the Delphi technique's outlines shown in Figure 1, and Table 5 summarizes the participants of the focus group discussion.

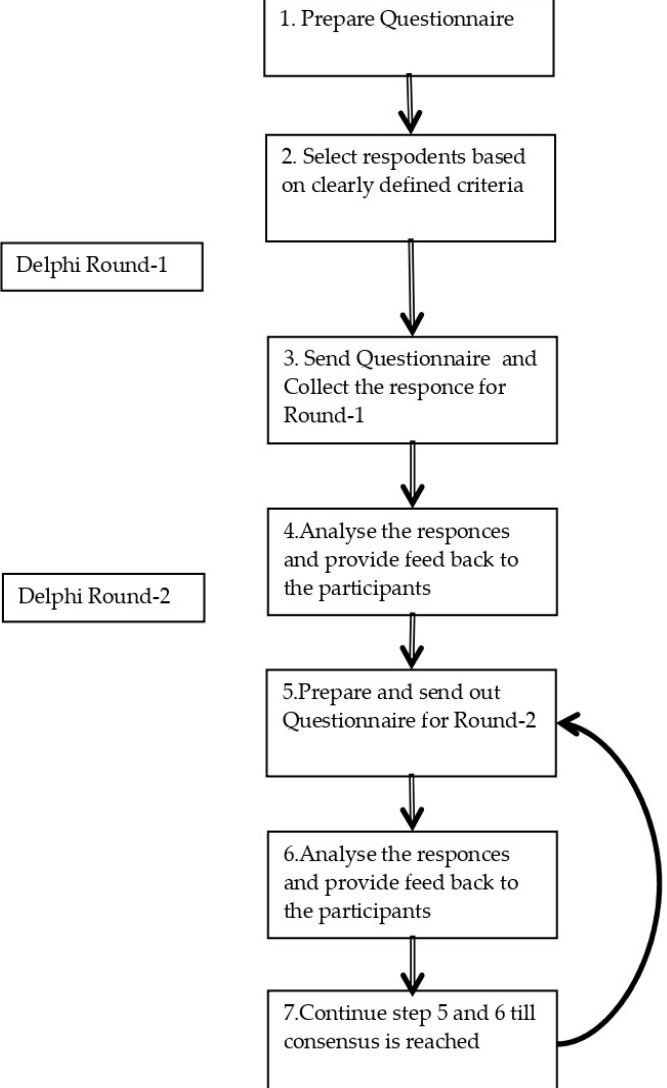

**Figure 1.** Overview of the Delphi technique.

These factors were presented to the field experts in round 1. After deliberations, six factors were removed, reducing them to 81. The revised questionnaire was presented again to experts in round 2, which resulted in further reduction of factors from 81 to 78 in accordance with relevance and objectives of the study. After round 2, the experts provided the following suggestions for the pilot study through the questionnaire.

1.  Participants of the pilot study must have more than 10 years relevant work experience in the field of study.

2.  The selected factors should be categorized as per the concept of environmental management study and life cycle assessment (LCA), and Figure 2 describes the accepted framework.

**Table 5.** Participants of focus group discussion.

| Designation | Qualification | Field Experience | Type of Organization |
|---|---|---|---|
| Director General | MS-Civil Eng. | 25 | Client |
| Director | BSc-Civil Eng. | 23 | Client |
| Director | BSc-Civil Eng. | 22 | Client |
| Deputy Director | BSc-Civil Eng. | 20 | Client |
| Deputy Director | MS-Civil Eng. | 20 | Client |
| Material Engineer | MS-Civil Eng. | 23 | Consultant |
| Project Manager | MS-Civil Eng. | 25 | Consultant |
| Construction Manager | MS-Civil Eng. | 22 | Consultant |
| Project Manager | BSc-Civil Eng. | 22 | Consultant |
| Owner/CEO | M.A | 25 | Contractor |
| Owner/CEO | BSc-Civil Eng. | 23 | Contractor |
| Owner/CEO | BSc-Civil Eng. | 20 | Contractor |

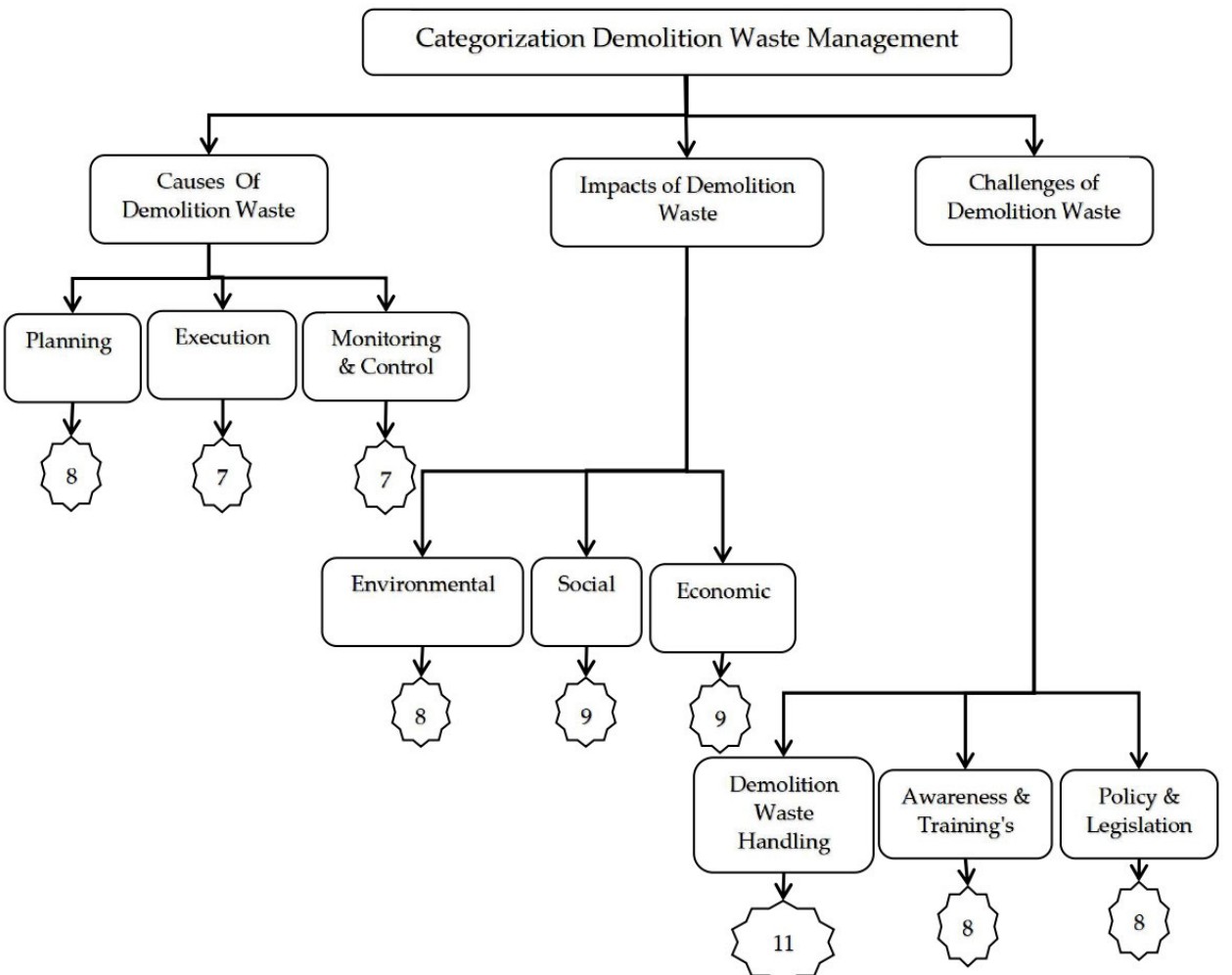

**Figure 2.** Categorization mechanism for demolition waste management (accepted framework).

For questionnaire survey to gather data the 5-point Likert scale was adopted, and the meaning of a specific point is as follows: "1—Very Low Impact; 2—Low Impact; 3—Moderate Impact; 4—High Impact; 5—Very High Impact".

*Pilot Study*

A pilot study is a preliminary investigation conducted on a smaller scale to gather logistical and methodological data relevant to a larger research project. The main purpose of a pilot study is to refine research instruments, procedures, and protocols, ensuring their validity and reliability before implementing them in the primary study. By conducting a pilot study, researchers can identify and address potential challenges, optimize study design, and improve the overall quality and effectiveness of the main study [71].

Twenty-seven participants were requested to participate in the pilot study, each having experience of more than twenty years in the construction sector. From the 27 participants, a 82% response rate was achieved. Table 6 shows the details of the participants involved in the pilot study.

**Table 6.** Participants of pilot study.

| Designation | Qualification | Field Experience | Type of Organization |
|---|---|---|---|
| Deputy Director | PhD Civil Eng | 26 | Client |
| Director | BSc-Civil Eng. | 21 | Client |
| Director | BSc-Civil Eng. | 20 | Client |
| Executive Engineer | BSc-Civil Eng. | 21 | Client |
| Superintendent Engineer | MS-Civil Eng. | 25 | Client |
| Assistant Director | BSc-Civil Eng. | 20 | Client |
| Executive Engineer | PhD Civil Eng | 24 | Client |
| Chief Eng | PhD Civil Eng | 27 | Client |
| Material Engineer | MS-Civil Eng. | 20 | Consultant |
| Material Engineer | MS-Civil Eng. | 20 | Consultant |
| Construction Manager | BSc-Civil Eng. | 25 | Consultant |
| Project Manager | MS-Civil Eng. | 25 | Consultant |
| Material Engineer | MS-Civil Eng. | 22 | Consultant |
| Material Engineer | MS-Civil Eng. | 23 | Consultant |
| Construction Manager | BSc-Civil Eng. | 22 | Consultant |
| Owner/CEO | DAE | 22 | Contractor |
| Owner/CEO | BSc-Civil Eng. | 25 | Contractor |
| Owner/CEO | BSc-Civil Eng. | 23 | Contractor |
| Owner/CEO | BSc-Civil Eng. | 25 | Contractor |
| Owner/CEO | DAE | 22 | Contractor |
| Owner/CEO | BSc-Civil Eng. | 25 | Contractor |
| Owner/CEO | DAE | 24 | Contractor |

The survey questionnaire was composed of 75 research items in total. Starting with seven questions framed around the general profiling of the respondents, they would have enabled the readers to understand the background of the targeted sample as well as their expertise in alignment with the opted research subject. This questionnaire was divided into two sections. Part one contained demographic information about the respondents, and part two included items to measure variables. Tables A1 and 7 show comprehensive details of the research model along with variable's code assigned to each factor in compliance with the framework (Figure 3).

**Table 7.** Variables coding.

| Section A | Sub-Section | Variable | Code |
|---|---|---|---|
| Causes of demolition waste generation | Planning stage | Discrepancies in bidding document | CW-PN-1 |
| | | Promoting sustainable material while designing | CW-PN-2 |
| | | Incorporation of demolition waste management in bidding process | CW-PN-3 |
| | | Incomplete bidding documents before tendering | CW-PN-4 |
| | | Scope/design changes | CW-PN-5 |
| | | Pre-demolition audit at planning stage | CW-PN-6 |
| | | Coordination and communication among stakeholders | CW-PN-7 |
| | | Implementation of lessons learnt | CW-PN-8 |
| | Execution stage | Inadequacy of implementation in waste management plan | CW-EX-1 |
| | | Role of supervision skills | CW-EX-2 |
| | | Errors and omissions in contract documents and need to revise contractual clauses | CW-EX-3 |
| | | Contract modification due to discrepancies | CW-EX-4 |
| | | Impact of eleventh-hour change of scope | CW-EX-5 |
| | | Consideration of site storage and space availability | CW-EX-6 |
| | | Waste generation due to poor workmanship | CW-EX-7 |
| | | Coordination and communication gap among stakeholders | CW-EX-8 |
| | Monitoring and control stage | Role of 3R (reduce, reuse, recycle) strategy | CW-MC-1 |
| | | Role of IT in demolition waste management mechanism | CW-MC-2 |
| | | Utilization of substandard materials resulting in wastage | CW-MC-3 |
| | | Inappropriate storage for unused construction materials | CW-MC-4 |
| | | Impact of on-site sorting techniques | CW-MC-5 |
| | | Impact of sub-letting/subcontracting | CW-MC-6 |
| | | Identification and quantification of demolition waste | CW-MC-7 |

| Section B | Sub-Section | Variables | Code |
|---|---|---|---|
| Demolition waste impacts | Environmental impacts | Green house gas emissions. | WI-EN-1 |
| | | Leaching effect (extraction of soluble chemical and mineral carriers into liquid by rainwater) e.g., acid | WI-EN-2 |
| | | Water contamination | WI-EN-3 |
| | | Air contamination due to pollution and dust generation | WI-EN-4 |
| | | Natural resources Consumption (i.e., construction material) | WI-EN-5 |
| | | Green land utilization and soil contamination | WI-EN-6 |
| | | Barrier to recharge for water table | WI-EN-7 |
| | | Indirect impact on creation of climate change (e.g., methane gas, etc.) | WI-EN-8 |
| | Social impacts | Health hazards in nearby communities. | WI-SO-1 |
| | | Additional human resource consumption | WI-SO-2 |
| | | Requirement of energy consumption | wI-SO-3 |
| | | Project stakeholders attitude towards DW management | WI-SO-4 |
| | | Sustainable development | WI-SO-5 |
| | | Vibrations and noise Pollution impacting society | WI-SO-6 |
| | | Health and safety impacts | WI-SO-7 |
| | | Lack of awareness of social impacts of demolition wastes | WI-SO-8 |
| | | Level of motivation for waste sorting/recycling/reusing | WI-SO-9 |
| | Economic impacts | Promotion and utilization of recycled materials | WI-EC-1 |
| | | Economic impact of additional incentive required for proper waste management | WI-EC-2 |
| | | Financial impact of recycling plants and stockpiled materials | WI-EC-3 |
| | | Impact of management and operation costs | WI-EC-4 |
| | | Costs associated with disposal of waste | WI-EC-5 |
| | | Financial aspect of creating awareness towards demolition management | WI-EC-6 |
| | | Limited knowledge of waste recycling/reuse | WI-EC-7 |
| | | Resources required for creating designated dumping zone | WI-EC-8 |
| | | Job creation opportunities | WI-EC-9 |

**Table 7.** *Cont.*

| Section C | Sub-Section | Variables | Code |
|---|---|---|---|
| Demolition waste management challenges | Demolition waste handling | Illegal dumping of demolition waste | WC-WH-1 |
| | | Strategic plans for effective demolition waste management | WC-WH-2 |
| | | Acceptance of demolition waste as recycling material by stakeholders | WC-WH-3 |
| | | Promotion of collaboration among stakeholders | WC-WH-4 |
| | | Arrangement for designated landfill sites | WC-WH-5 |
| | | Role of regulatory control and government legal enforcement | WC-WH-6 |
| | | Issues With supply chain management | WC-WH-7 |
| | | Industry support for effective utilization of demolition waste | WC-WH-8 |
| | | Limited specialized demolition waste handling companies | WC-WH-9 |
| | | Demolition waste management budget allocation | WC-WH-10 |
| | | Level of support by government agencies for environmental protection promotion | WC-WH-11 |
| | Awareness and training | Knowledge of supplier's co-operation towards waste material utilization | WC-AT-1 |
| | | Role of project executing staff towards demolition waste management | WC-AT-2 |
| | | Support of existing practice and policies | WC-AT-3 |
| | | Role of technological support for smart construction | WC-AT-4 |
| | | Arrangement of in-house training on environmental management | WC-AT-5 |
| | | Role of awareness and training for environmental protection | WC-AT-6 |
| | | Impact of industrial focus on cost and time rather than demolition waste management | WC-AT-7 |
| | | Level of demolition waste management by trained staff and expert personnel as per contract document | WC-AT-8 |
| | Policy and legislation | Inadequacy of industry norms | WC-PL-1 |
| | | Engaging all types of social media for promoting demolition waste management | WC-PL-2 |
| | | Demolition waste management consideration in project life cycle | WC-PL-3 |
| | | Level of enforcement of waste management plan | WC-PL-4 |
| | | Penalization mechanism for generating demolition waste and damaging environment | WC-PL-5 |
| | | Role of legislation for enforcing policies to protect environment | WC-PL-6 |
| | | Enforcement of legal requirements on environmental protection | WC-PL-7 |
| | | Industry culture of incentive policies to promote utilization of demolition waste management | WC-PL-8 |

Based upon the frequency analysis (Delphi Rounds 1 and 2) and pilot study, finally, 75 factors have been selected for the development of the final questionnaire. It is ensured that the participants involved in the Delphi Rounds 1and 2 and pilot study are not the same. The conceptual framework for the research study is presented in Figure 3.

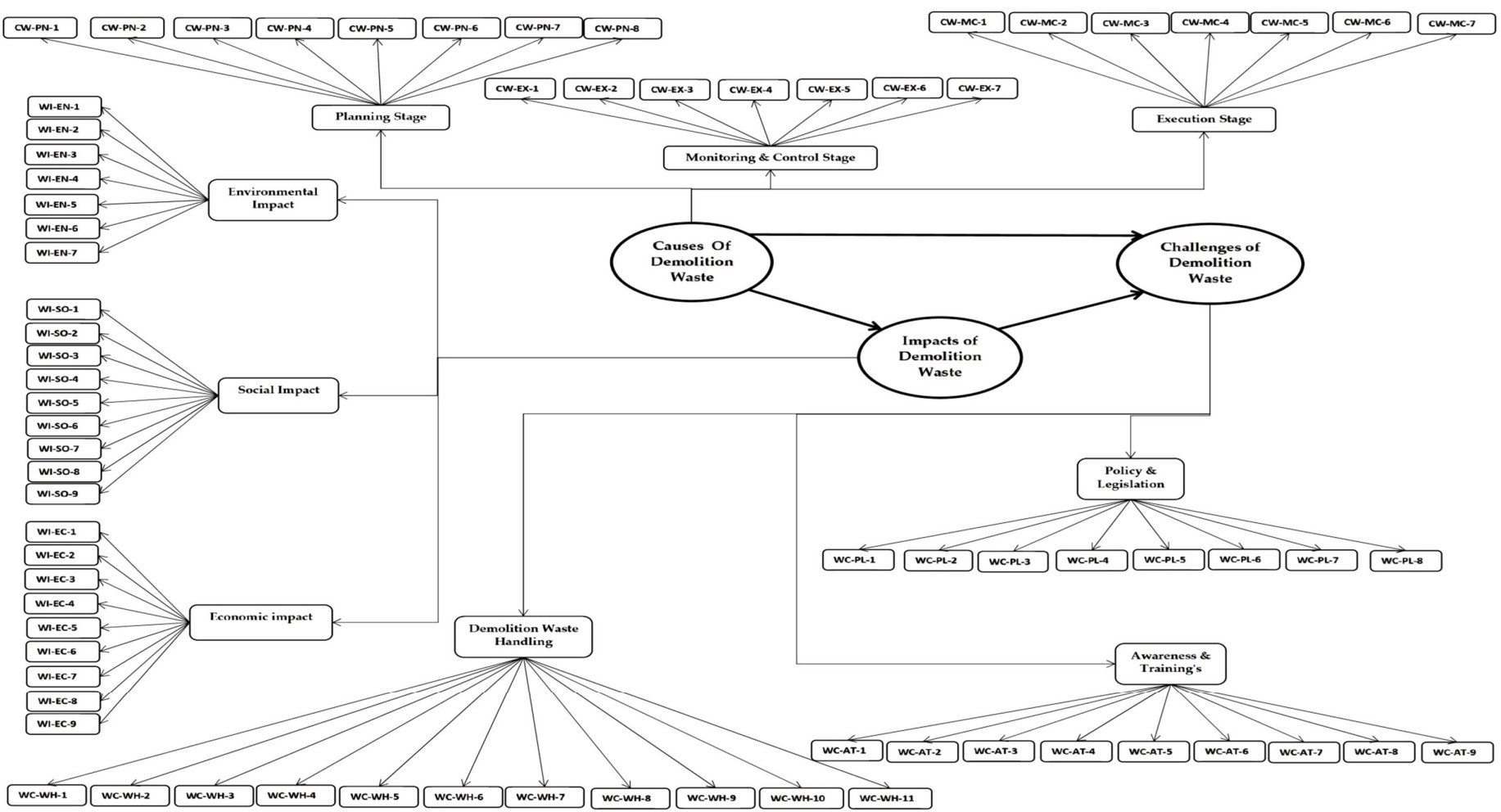

**Figure 3.** Conceptual framework for research study using Partial Least Squares Structural Equation Modeling (PLS-SEM).

This approach is widely utilized in various disciplines including social sciences, business, and economics to model and uncover relationships between latent variables, offering insights into complex interrelationships within datasets despite inherent limitations in sample size or collinearity among variables. Its flexibility and robustness make PLS-SEM a valuable tool for analyzing and understanding intricate relationships in research contexts such as construction and demolition waste management.

Choosing Partial Least Squares Structural Equation Modeling (PLS-SEM) as the methodological framework for studying demolition waste management is motivated by several factors. Firstly, the complexity of variables involved in waste management, such as waste generation rates and recycling efficiency, can be effectively handled by PLS-SEM, which captures relationships between latent constructs and observed variables. Secondly, PLS-SEM's robustness with limited sample sizes makes it suitable for studies focusing on specific regions or timeframes where data may be scarce. Thirdly, the method's flexibility with non-normally distributed data aligns with the diverse sources often encountered in waste management research. Additionally, PLS-SEM's ability to model formative constructs is advantageous for capturing variables like "waste management practices" or "sustainability initiatives." Its predictive modeling capability supports forecasting waste management outcomes and evaluating strategy effectiveness. Moreover, PLS-SEM can integrate stakeholder perspectives and allows iterative model refinement, crucial in dynamic environments like waste management. Lastly, its clear and interpretable results enhance accessibility for stakeholders involved in decision making. Overall, employing PLS-SEM facilitates the comprehensive understanding of demolition waste management dynamics, fostering sustainable and efficient practices [88]. Research model for this research based on PLS-SEM is shown below in Figure 4

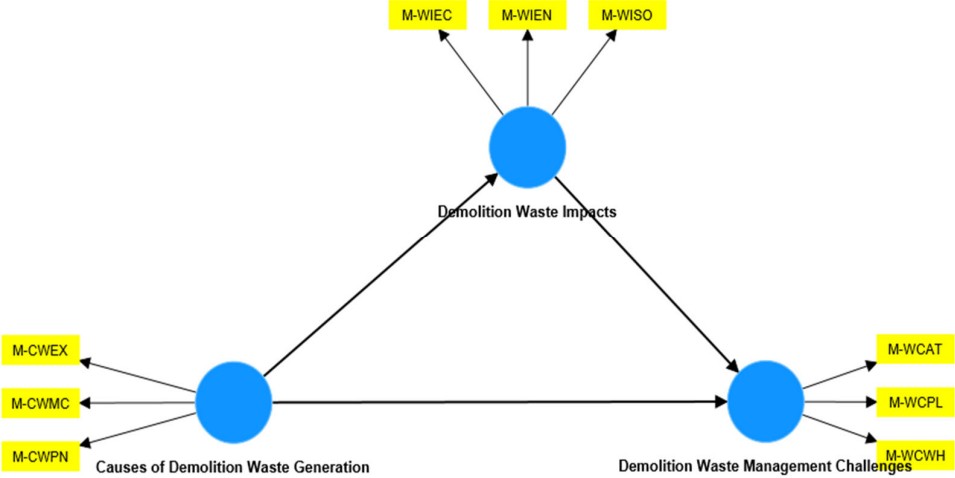

**Figure 4.** Research model using PLS-SEM.

Connecting proposed code categories with Partial Least Squares Structural Equation Modeling (PLS-SEM) involves integrating qualitative data analysis with quantitative modeling. This integration begins by treating each code category developed through qualitative analysis as a latent variable in the PLS-SEM structural model. These variables represent constructs or dimensions relevant to the research question. The relationships between categories and other variables in the model are then specified, guided by theoretical grounding and informed by the qualitative analysis. Quantitative measures derived from the coded data serve as indicators for these latent variables, providing a quantitative representation of the underlying constructs [89].

Descriptive analysis of categorical items was conducted to assess the normality of the data. Mean, standard deviation, skewness, and kurtosis were examined for this purpose. Following this, factor analyses were performed to ascertain the reliability and validity of utilizing these factors in measurement models for assessing public housing performance.

These analyses encompassed reliability assessments, exploratory factor analyses, and confirmatory factor analyses. Figure 5 shows the sequences of the PLS-SEM approach.

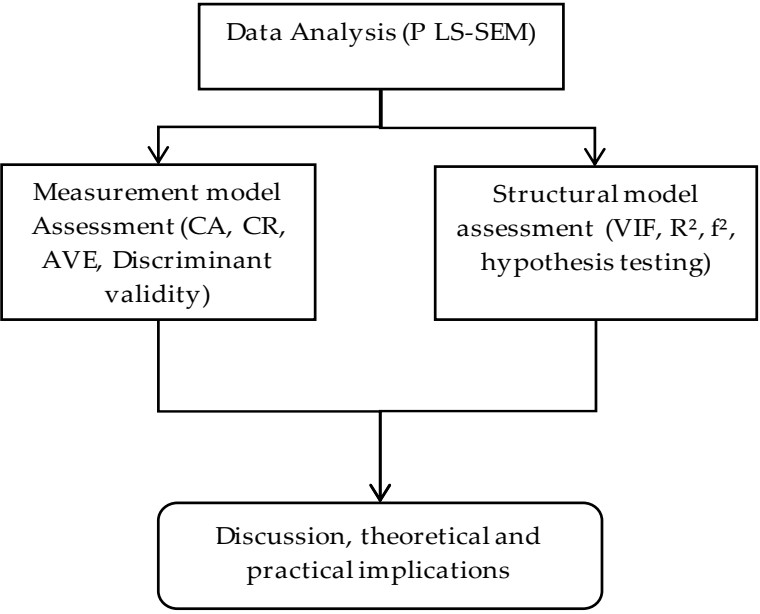

**Figure 5.** Flow chart of PLS-SEM.

This structured approach provides a clear and comprehensive framework for reporting the results of a Partial Least Squares Structural Equation Modeling (PLS-SEM) analysis in academic or professional contexts. By systematically addressing model fit and assessment, measurement model results, structural model results, bootstrapping results, overall model evaluation, and optional visualization, researchers can effectively communicate the findings of their analysis while adhering to standards of rigor and clarity. This approach ensures that the key aspects of the analysis, such as model validity, reliability, significance of relationships, and practical implications, are thoroughly addressed and documented, contributing to the advancement of knowledge in the field. Comprehensive overview of the statistical techniques and assessment criteria commonly used in Partial Least Squares Structural Equation Modeling (PLS-SEM) to evaluate model fit, reliability, validity, and other key aspects are shown below in Table 8.

**Table 8.** Statistical techniques.

| Aspect | Techniques/Criteria | Description | Range/Threshold |
|---|---|---|---|
| Model fit and assessment | Goodness-of-fit measures (e.g., $R^2$, $Q^2$) | Provide insights into the overall fit of the model. Higher values generally indicate better fit, but interpretation varies. | There is no universal threshold. |
| Measurement model assessment | Reliability: Cronbach's alpha Composite reliability (CR)—Rho_A | Reliability measures are used to assess the internal consistency of scales or constructs. | Values expected to exceed 0.60. |
| | Convergent validity: Factor loadings | Factor loadings indicate the strength of relationships between observed variables and their underlying constructs. | Ideally, factor loadings should exceed 0.60. |
| | Discriminant validity: comparison of square roots of AVE to interconstruct correlations | Ensures that each construct is distinct from others. Square roots of AVE (Average Variance Extracted) should be greater than the correlations between the constructs. | |
| Structural model assessment | Path coefficients: significance and direction evaluation $R^2$ | Evaluate significance and direction of relationships between constructs. | $R^2$ values represent the proportion of variance explained by endogenous constructs. |

**Table 8.** *Cont.*

| Aspect | Techniques/Criteria | Description | Range/Threshold |
|---|---|---|---|
| Bootstrapping | Bootstrap re-sampling: significance testing of path coefficients Estimation of confidence intervals | Used for significance testing of path coefficients and estimation of confidence intervals. | Bootstrap ratios (t-values) are assessed against critical values to determine significance. |
| Multicollinearity assessment | Variance Inflation Factor (VIF) | Measures the degree of multicollinearity among predictors. | Values ideally less than 5 or 10, though some researchers suggest a more stringent cut-off of 3. |
| Cronbach's alpha | Reliability measure | Assesses internal consistency of scales or constructs. | Values expected to exceed 0.70, though higher values are preferable. |
| Interpretation | Contextual interpretation: Research question Theoretical framework Practical implications of the study | Essential to interpret results in the context of specific research questions, theoretical frameworks, and practical implications of the study. | |

**Hypothesis 1:** *There is a significant impact of causes of demolition waste generation on demolition waste impacts.*

This hypothesis is grounded in the understanding that the causes of demolition waste generation, such as construction practices, material choices, and project management decisions, can have significant implications for the environmental, social, and economic impacts of demolition activities. For example, inefficient construction practices, poor waste segregation, and lack of recycling infrastructure can lead to higher waste generation rates and increased environmental degradation. Theoretical frameworks such as life cycle assessment (LCA) and environmental impact assessment (EIA) provide a basis for understanding the relationships between demolition waste generation causes and their impacts.

Research studies utilizing LCA, EIA, and other impact assessment methodologies have documented the environmental, social, and economic impacts of demolition waste generation. These studies often analyze the environmental footprint of demolition activities, including greenhouse gas emissions, energy consumption, air and water pollution, and habitat destruction. Additionally, case studies and industry reports provide evidence of specific causes of demolition waste generation, such as building design, construction methods, material selection, and project management practices, and their associated impacts on waste generation and disposal. The hypothesis-1 of this study was tested using PLS-SEM and is shown in Figure 6.

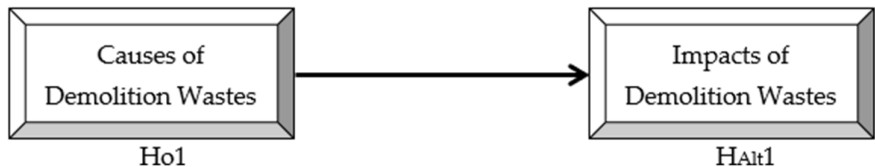

**Figure 6.** Hypothesis-1 for demolition waste management for this study.

**Hypothesis 2:** *There is a significant impact of causes of demolition waste generation on demolition waste management challenges.*

This hypothesis is based on the recognition that the causes of demolition waste generation can create significant challenges for waste management practices and infrastructure. Factors such as the heterogeneity of demolition waste streams, contamination levels, and regulatory requirements can complicate waste sorting, recycling, and disposal processes. Theoretical perspectives from waste management and environmental governance highlight the importance of addressing upstream factors that influence waste generation to effectively manage waste downstream.

Research studies and industry reports have identified various challenges associated with the causes of demolition waste generation. These challenges include inadequate waste characterization and sorting facilities, limited recycling capacity, lack of market demand for recycled materials, and regulatory barriers to waste diversion and recycling. Case studies and best practice guides provide evidence of successful strategies for overcoming these challenges, such as improved waste sorting techniques, investment in recycling infrastructure, and policy incentives for sustainable demolition practices. The hypothesis-2 of this study was tested using PLS-SEM and is shown in Figure 7.

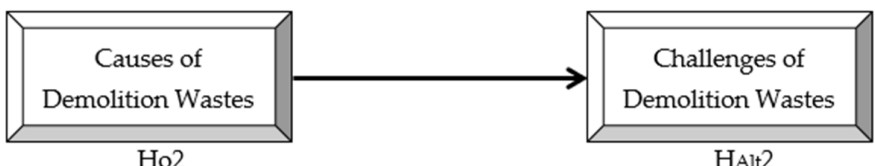

**Figure 7.** Hypothesis-2 for demolition waste management for this study.

**Hypothesis 3:** *There is a significant impact of demolition waste impacts on demolition waste management challenges.*

This hypothesis posits that the environmental, social, and economic impacts of demolition waste, such as pollution, habitat destruction, public health risks, and economic costs, can exacerbate existing challenges in waste management practices. Theoretical perspectives from environmental sociology, risk assessment, and policy studies emphasize the interconnectedness of environmental problems and the need for integrated approaches to address complex challenges.

Research studies and policy reports have documented the impacts of demolition waste on waste management challenges. These impacts may include increased landfill pressure, regulatory compliance costs, public opposition to waste facilities, and health and safety risks for waste workers and surrounding communities. Case studies and empirical research provide evidence of the linkages between demolition waste impacts and waste management challenges, highlighting the need for holistic and proactive approaches to waste management and environmental protection. The hypothesis-3 of this study was tested using PLS-SEM and is shown in Figure 8.

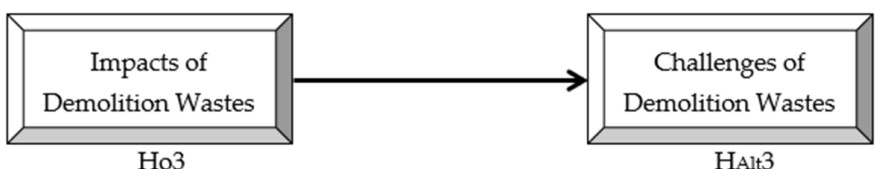

**Figure 8.** Hypothesis-3 for demolition waste management for this study.

The current study opted for quantitative cross-sectional research approach obtained the information from the relevant respondents and made deductions with regard to the variables under consideration. The population selected for the current study consisted of specialized professionals working in Pakistan's infrastructural development sector who either directly have been managing the demolition waste management in their current or past roles or are at least familiar with its concepts. Therefore, sampling for this study is performed using the purposive approach, as it involved reaching out to experts in the field of demolition waste management across Pakistan. This approach enabled the collection of precise data points that were relevant to the current study and mitigate the risk of irrelevant data inclusion and cost overflow and time and resources needed to collect the data. Purposive sampling, also referred to as judgmental or selective sampling, is a qualitative research method where participants are deliberately chosen based on specific criteria relevant to the research objectives, rather than being randomly selected.

Unlike probability sampling methods that aim to give every member of the population an equal chance of selection, purposive sampling relies on the researcher's discretion in selecting participants.

In qualitative research, purposive sampling is commonly used to gain detailed insights, investigate specific phenomena, or understand particular perspectives within a specific population. Determining the sample size for purposive sampling differs from probability sampling methods because it is not based on statistical formulas. Instead, the sample size is determined by factors such as research goals, population characteristics, available resources, and the principle of data saturation.

Key considerations for determining sample size in purposive sampling include the following:

Research Objectives: Clearly define the study's objectives and ensure the sample size is sufficient to address these objectives effectively.

Expertise and Judgment: Rely on the expertise and judgment of researchers familiar with the field to determine an appropriate sample size based on the research question and context.

There is no fixed rule for determining sample size in purposive sampling; it requires the thoughtful consideration of various factors and may need adjustments based on the specific context and goals of the study.

## 3. Results

Conclusively, 150 questionnaires were sent out to the targeted population of professionals working in the construction sector. At the conclusion of the survey process, a total of 115 responses were gathered, out of which 21 were discarded based on the premise of incomplete information or incorrect/redundant data. Therefore, a response rate of 81.7% was achieved, and utilizable data was obtained from 94 participants.

To begin with, the respondents were classified in terms of their current role as well as their expertise in demolition waste management. Considering this, the first attribute referred to the classification based on the type of organizations, for which 53 firms were based on the ownership/client model, 23 firms involved the consultancy-based model, and 18 served as contractors, while 0 were attributed to the remaining possible categories. Secondly, in terms of job roles, 27 respondents were serving in capacity of a CEO/MD, 14 as a project manager, 8 as a project architect, 9 as a project quantity surveyor, and 36 as a project engineer, while 0 were employed in other roles in the infrastructure development sector. Thirdly, the next stage of profiling involved categorizing the respondents in terms of their overall experience in the industry. In this regard, 0 participants held less than 5 years of relevant industry experience, 0 were in the construction sector for 6–10 years, and 10 were serving for 10–15 years, while 26 held a tenure record of 15–20 years. Though 58 respondents were the most experienced, they had served the construction industry for over 20 years. Fourthly, as far as the nature of projects involved in the current research is concerned, 44 were based on traditional contracts, 32 were based on design and build approach, 13 were turnkey, and 5 were based upon management contracts, while 0 were based upon other approaches. Furthermore, the respondents were questioned about their familiarity with regard to the demolition waste management, as response to which 94 responded a 'yes', while 0 had no first-hand experience in managing demolition waste. In alignment to this, the respondents were questioned about the inclusion of reduce and reuse practices as a part of their assigned projects, for which 94 did confirm the observance of aforementioned practices, while 0 negated the inclusion of the said approach. Lastly, the participants were questioned regarding their own experience-based personal opinion on demolition waste management plan being a part of construction contracts that are signed between the involved stakeholders. In their responses, 94 indicated it to be a crucial consideration, while 0 considered it to be unnecessary.

The demographic distribution of the respondents for this study is illustrated in Figure 9. It highlights the experience of the respondents, their type of organization, designation, and project that they handled.

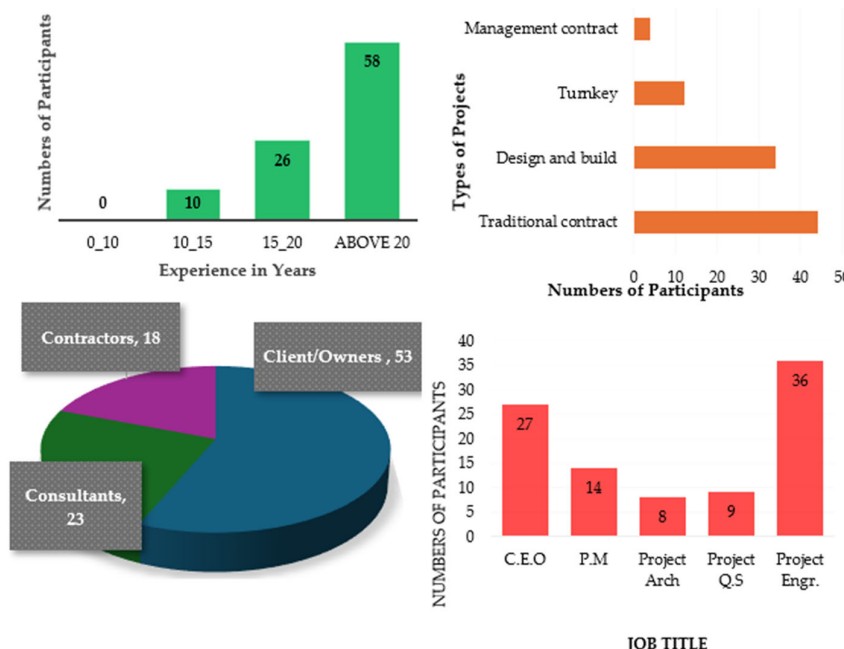

**Figure 9.** Demographic classification.

In the next stage, the research items were tested in alignment with their attributed research variables as well as the proposed hypothesis. Firstly, the opted research variables were tested for their internal variance as well as the correlatability among the respective research items, in terms of factor loadings. Keeping that in mind, ref. [90] suggested that each research item should be valued equal or above the threshold of 0.6. As result, it was observed that each of the variables had a factor loading valued well above 0.6. Therefore, they were deemed to be fit for further testing, without removing any of the research items from the devised model (See Table 9).

**Table 9.** Outer loadings.

| Items | Causes of Demolition Waste Generation | Demolition Waste Impacts | Demolition Waste Management Challenges |
|---|---|---|---|
| M-CWEX | 0.948 | | |
| M-CWMC | 0.930 | | |
| M-CWPN | 0.921 | | |
| M-WCAT | | | 0.955 |
| M-WCPL | | | 0.979 |
| M-WCWH | | | 0.736 |
| M-WIEC | | 0.904 | |
| M-WIEN | | 0.911 | |
| M-WISO | | 0.913 | |

Secondly, the statistical testing involved testing the research instrument for their reliability to generate accurate results in terms of measuring the respective phenomena associated to each variable, regardless of the environment that they are being testing in. Considering which, Cronbach's alpha is considered as a widely accepted parameter to gauge the reliability of opted research instrument [91,92]. Considering the minimum threshold for the said parameter to be 0.6, the research variables performed really well with causes of demolition waste generation valued at 0.62 and demolition waste impacts valued at 0.89, and demolition waste management challenges received a reliability score of 0.87.

Further, a research instrument's validity suggests the selection of relevant items to measure a phenomenon. Keeping that in mind, the validity is categorized as convergent validity and discriminant validity. The former one suggests measuring the internal relevance of research items of any given variable with one another, therefore assuring that each of them shares common underlying theories in terms of defining a variable. To gauge the said phenomena, SEM offers AVE (Average Variance Extracted) as a parameter to evaluate the convergent validity of any given research instrument. Conclusively, for the current study, all the valuables were valued well above the minimum threshold of 0.5 recommended for AVE [91,93]. Therefore, all the adapted research items were deemed to be convergently valid (See Table 10).

**Table 10.** Instrument reliability.

| | Cronbach's Alpha | Composite Reliability (rho_a) | Composite Reliability (rho_c) | Average Variance Extracted (AVE) |
|---|---|---|---|---|
| Causes of demolition waste generation | 0.926 | 0.930 | 0.953 | 0.870 |
| Demolition waste impacts | 0.896 | 0.915 | 0.935 | 0.827 |
| Demolition waste management challenges | 0.874 | 0.935 | 0.924 | 0.804 |

Next, the discriminant validity conforms to the dissimilarity existent between the opted research items of each variable from the research items of other variables in the study. This assures that this is no overlap or redundancy among the research items gauging any given research variable. In relevance to the SEM approach, cross-loading is one of the parameters to measure the discriminant validity of each respective research item individually. For each item to be discriminately valid, it is required to have a higher correlational value with its own corresponding item, in comparison to rest of the variables in the study [94]. Keeping that in mind, all research items were found to be valid in the current research (See Table 11). Research model along with loading is shown in Figure 10.

**Table 11.** Cross loadings.

| Items | Causes of Demolition Waste Generation | Demolition Waste Impacts | Demolition Waste Management Challenges |
|---|---|---|---|
| M-CWEX | 0.948 | 0.736 | 0.821 |
| M-CWMC | 0.930 | 0.868 | 0.881 |
| M-CWPN | 0.921 | 0.736 | 0.792 |
| M-WCAT | 0.855 | 0.901 | 0.955 |
| M-WCPL | 0.933 | 0.939 | 0.979 |
| M-WCWH | 0.560 | 0.585 | 0.736 |
| M-WIEC | 0.875 | 0.904 | 0.969 |
| M-WIEN | 0.635 | 0.911 | 0.743 |
| M-WISO | 0.745 | 0.913 | 0.762 |

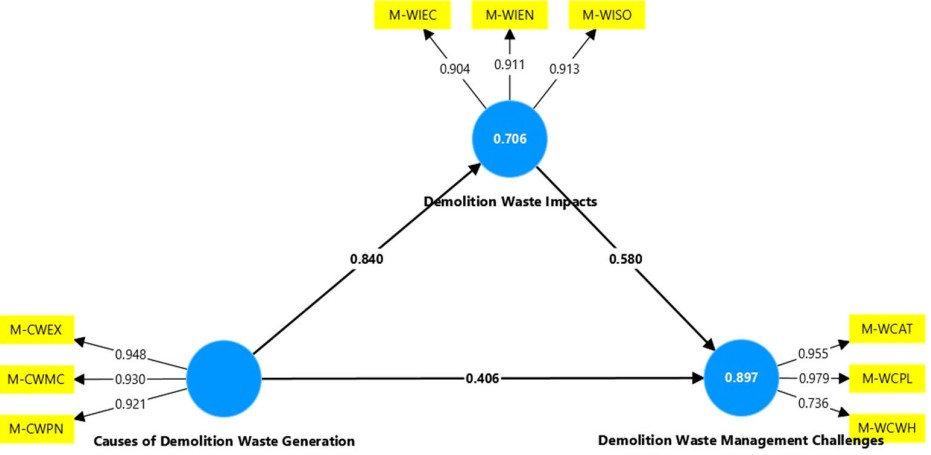

**Figure 10.** Research model using PLS-SEM with loading.

In the SEM, another parameter is Fronell–Larcker Criterion, which evaluates discriminant validity by comparing the AVE of each latent variable with its correlations with other latent variables in the model. As per the given criterion, a variable is required to have a high correlational value with itself rather than the rest of variables in the study [94]. As a result, it was observed that each variable fulfilled the said criterion (See Table 12).

**Table 12.** Fronell–Larcker Criterion.

| | Causes of Demolition Waste Generation | Demolition Waste Impacts | Demolition Waste Management Challenges |
|---|---|---|---|
| Causes of demolition waste generation | 0.933 | | |
| Demolition waste impacts | 0.840 | 0.909 | |
| Demolition waste management challenges | 0.893 | 0.901 | 0.897 |

Finally, in terms of assessing discriminant validity, the higher level of specificity is typically valued between 97% and 99%, and it likely refers to the threshold or criterion used for the HTMT ratio to ensure strong discriminant validity. HTMT ratio is used to assess discriminant validity, and it is essential to aim for values well below 1 (typically in the range of 0.97 to 0.99) to ensure a high level of confidence in the distinction between constructs. This supports the validity of the research instrument by demonstrating that the measured constructs are distinct and not overly related to each other. Considering this criterion, the values associated with all variables in the present research were below 1 [92,93]. Therefore, the research instrument is deemed to be discriminately valid (See Table 13).

**Table 13.** HTMT—Heterotrait–Monotrait Ratio.

| | Causes of Demolition Waste Generation | Demolition Waste Impacts | Demolition Waste Management Challenges |
|---|---|---|---|
| Causes of demolition waste generation | | | |
| Demolition waste impacts | 0.902 | | |
| Demolition waste management challenges | 0.970 | 0.906 | |

Once the reliability and validity of the research instrument are verified, the next stage involves evaluating the research items for their internal consistency in terms of the variance inflation factor (VIF). The VIF (Variance Inflation Factor) indicates a high level of correlation between the variables and their respective indicators. The said measure requires each research items to be valued under five, in order to be considered a fit [94]. Keeping that in mind, all the research items of the present research were found to fit (See Table 14).

**Table 14.** Variance Inflation Factor (VIF).

| Item | VIF |
|---|---|
| M-CWEX | 4.602 |
| M-CWMC | 3.307 |
| M-CWPN | 3.455 |
| M-WCAT | 4.512 |
| M-WCPL | 17.945 |
| M-WCWH | 1.756 |
| M-WIEC | 2.244 |
| M-WIEN | 3.301 |
| M-WISO | 3.226 |

Following this, to determine the effect size of each variable (f2), an indicator representing the magnitude of effect an exogenous variable may have on an endogenous research variable is utilized. The magnitude of the effect is categorized into three different ranges. Firstly, if the values of effect size (f2) fall between 0.02 and 0.14, it is considered a small effect. Further, for relational effects ranging between 0.15 and 0.35, they are considered medium effects. Lastly, values equal to or above 0.36 are classified as large effects [92,93]. In reference to the given ranges, causes of demolition waste generation, demolition waste

impacts, and demolition waste management challenges, all three were valued as ones with high impact (See Table 15).

**Table 15.** Effect size (f2).

| | Causes of Demolition Waste Generation | Demolition Waste Impacts | Demolition Waste Management Challenges |
|---|---|---|---|
| Causes of demolition waste generation | | 2.399 | 0.472 |
| Demolition waste impacts | | | 0.965 |
| Demolition waste management challenges | | | |

In order to determine the overall predictability/impact percentage of endogenous variables to determine the endogenous variable, the parameter of coefficient of determination ($R^2$) is utilized [92,93]. Considering that the causes of demolition waste generation and demolition waste impacts contributed to 89.7%, the demolition waste management challenges in the construction sector were determined (See Table 16).

**Table 16.** Coefficient of determination ($R^2$).

| | R-Square | R-Square Adjusted |
|---|---|---|
| Demolition waste impacts | 0.706 | 0.703 |
| Demolition waste management challenges | 0.897 | 0.895 |

Conclusive in determining that the current research model fulfilled the fitness criterion of the measurement model, the calculations regarding the structural model enabled determining the impact of one variable on another, as this assisted in gauging the most impactful research variable in the current research. Considering which, path coefficient is an effective parameter to evaluate the impact magnitude. It ranges from $-1$ to $+1$, reflecting a maximum negative or positive impact attributed to a variable. Further, the significance of each individual impact is determined by it associated p-value being under 0.05 [92,93]. Keeping that in mind, causes of demolition waste generation depicted the most impact on demolition waste impacts, with an evaluated path coefficient value of 0.840. Following which, impact of demolition waste impacts on determining the demolition waste management challenges was valued at 0.580, in terms of path coefficient. Lastly, causes of demolition waste generation showed least impact on demolition waste management challenges with a path coefficient valued at 0.406. For all the aforementioned relationships, the respective *p*-value was well under the threshold of 0.05. Therefore, deeming all the evaluated relationships as significant (See Table 17).

**Table 17.** Path coefficients.

| | Original Sample (O) | Sample Mean (M) | Standard Deviation (STDEV) | T Statistics (\|O/STDEV\|) | *p* Values | Accepted/Rejected |
|---|---|---|---|---|---|---|
| Causes of demolition waste generation -> demolition waste impacts (Ho1-HAlt1) | 0.840 | 0.843 | 0.016 | 52.897 | 0.000 | Accepted |
| Causes of demolition waste generation -> demolition waste management challenges (Ho2-HAlt2) | 0.406 | 0.408 | 0.043 | 9.376 | 0.000 | Accepted |
| Demolition waste impacts -> demolition waste management challenges (Ho3-HAlt3) | 0.580 | 0.578 | 0.043 | 13.467 | 0.000 | Accepted |

Finally, the mediatory effect of demolition waste between causes of demolition waste generation and demolition waste management challenges was measured by calculating the specific indirect impact between the variables. Keeping that in mind, the path coefficient was found to be significant and was valued at 0.488. Therefore, demolition waste impacts were deemed as an effective mediator to be considered and explained the impact between the aforementioned variables [92,93] (See Table 18).

**Table 18.** Mediation effect.

| | Original Sample (O) | Sample Mean (M) | Standard Deviation (STDEV) | T Statistics (|O/STDEV|) |
|---|---|---|---|---|
| Causes of demolition waste generation -> demolition waste management challenges | 0.488 | 0.487 | 0.035 | 13.779 |

Questionnaire survey-based methodology along with application of PLS-SEM assessed the interrelation among the adopted variables. The study concluded the following:

- The measurement instruments used to assess causes of demolition waste generation, demolition waste impacts, and demolition waste management challenges are reliable and valid for the study's purposes. Overall, the measures suggest that the constructs in the study have high reliability and validity.
- Higher factor loadings suggested stronger relationships, indicating the carriable items, considered in the study, present good indicators of their respective constructs.
- Achieving a higher AVE than its correlations, each construct indicated a distinct and unique identity, thus supporting the fact that the constructs are measuring different aspects of the phenomenon under investigation. These correlations represented the degree of association between different constructs, and the higher its values, the stronger associations between the constructs.
- Besides many other variables, "M-WCPL" was the only item that had a notably high VIF value of 17.945, which may indicate multicollinearity issues with other variables. Thus, further investigation is advised to understand the relationships between variables, especially involving "M-WCPL," to assess and address potential multicollinearity concerns.
- In terms of path coefficients, overall, the results suggest that all three hypothesized relationships are statistically significant in our model. These coefficients indicate the strength and direction of the relationships between the constructs. Statistical information such as T statistics and P values helps to determine the significance of these relationships. In all cases, the relationships are found to be statistically significant (*p* values = 0.000) and are accepted.

Overall, based on results, it is observed that "Causes of Demolition Waste Generation" has a significant mediation effect on "Demolition Waste Management Challenges". This study found that in the three hypothesis, Ho1-HAlt1, Ho2-HAlt2, and Ho3-HAlt3, for all the aforementioned relationships, the respective p-value was well under the threshold of 0.05. Therefore, all the evaluated relationships were deemed as significant, and this investigation confirmed that all hypotheses were supported.

### 3.1. Novelty of the Study—Partial Least Squares Structural Equation Modeling (PLS-SEM)

The current work presents the novelty in the following terms:

- Based on the critical literature review, there exists a research gap for environmental management specific to road demolition waste. This becomes more relevant in the case of developing countries having rapid urbanization and population growth where road network is key to the social and as the economic upliftment.
- The comprehensive management of all phases of the life cycle seems to be missing.
- A numerical correlation between various variables responsible for environmental management for infrastructure facilities, specifically road network, to support the sustainable demolition waste management practices is also missing.

### 3.2. The Benefits of the Study

Managing demolition waste effectively offers several benefits like economic, pollution minimization, etc. and are as follows:

3.2.1. Economics View

The economic benefits of managing demolition waste from roads stem from cost savings, revenue generation, resource conservation, compliance with regulations, and the promotion of corporate social responsibility. By adopting sustainable waste management practices, construction companies can optimize economic efficiency, reduce environmental impact, and contribute to the long-term viability of infrastructure projects. From this study, the benefits from the economic point of view are as follows:

- Proper management of road demolition waste can help reduce disposal costs associated with landfilling or incineration, thus avoiding disposal expenditures.
- Effective management of road demolition waste ensures compliance with environmental regulations and standards, avoiding potential fines or penalties for non-compliance. By adhering to waste management regulations, construction companies can avoid costly legal disputes and reputational damage.
- Implementing sustainable waste management practices for road demolition waste demonstrates a commitment to corporate social responsibility (CSR). This can enhance the reputation of construction companies, attract environmentally conscious clients, and create opportunities for partnerships and collaborations. Thus, supporting a healthy and sustainable economic competition.
- This study will guide the conservation of valuable resources and minimize the environmental impact of resource extraction, contributing to long-term economic sustainability.

3.2.2. Pollution Minimization

Currently, the study is in its initial phase. As per authors' understanding, the developed system for managing demolition waste from roads can serve as a model or framework that can be support other cases within the construction industry or beyond with some modifications according to their circumstances. However, the proposed framework may not be able to address all aspects of pollution minimization comprehensively on its own. It can only provide a solid foundation for promoting sustainable practices and reducing environmental impact. By applying the principles of waste reduction, recycling, pollution prevention, and continuous improvement, the system can contribute to broader efforts to minimize pollution and promote environmental sustainability across various sectors and contexts.

**4. Conclusions**

The current study, Environmental Management Framework for Road Network Demolition Wastes, explored the mediating role of demolition waste management in the relationship among the causes of demolition waste generation, the impacts of demolition waste, and the challenges of demolition waste management.

This study examined the facilitating role of improvement factors on demolition waste management, which in turn significantly impact sustainable demolition waste management, thereby enriching the existing body of knowledge. Additionally, it offers a practical contribution by proposing a strategic methodological approach to assist management in enhancing the performance of small businesses and ensuring their long-term effectiveness.

**5. Recommendations**

The study proposed an insightful approach by providing a framework for evaluating the influence of three main sections for DWM. Additionally, it develops a novel integrated framework aimed at assisting policymakers and construction industry professionals in reducing demolition waste and improving waste management practices. Based on the findings of this study, the recommendations are as follows:

- Integration of sustainable construction practices into the curriculum of educational institutions through proper research endeavors, along with inclusion in professional development programs for individuals in the construction industry, would greatly enhance the concept of managing demolition wastes in a more sustainable manner.

- The output of study can help to develop specific legislation to govern the handling and disposal of demolition wastes, accompanied by strict monitoring mechanisms to ensure compliance. This enhances the efficiency and effectiveness of demolition waste management, leading to better environmental and societal outcomes. More emphasis needs to be given to a solution-oriented approach in line with the explored areas of demolition waste life cycle.

- According to the model, the components of the demolition waste management model may suggest ways to handle demolition waste more sustainably. Future research should concentrate on global construction industry standards, waste management practices, and mechanisms for segregating construction and demolition debris from municipal waste streams.

**Author Contributions:** Conceptualization, S.S.S.G.; Validation, S.S.S.G.; Investigation, I.H.; Resources, I.H.; Data curation, S.S.U.; Writing—original draft, S.S.U.; Writing—review & editing, S.S.S.G. All authors have read and agreed to the published version of the manuscript.

**Funding:** This research received no external funding.

**Institutional Review Board Statement:** Not applicable.

**Informed Consent Statement:** Consent was obtained from professional for data collection for all subjects involved in the study.

**Data Availability Statement:** The data may be available on request.

**Conflicts of Interest:** The authors declare no conflict of interest.

## Appendix A

**Table A1.** Coding guidelines.

| Factors | Code |
| --- | --- |
| Causes of demolition waste generation | CW |
| Causes of demolition waste generation at planning stage | CW-PN |
| Causes of demolition waste generation at execution stage | CW-EX |
| Causes of demolition waste generation at monitoring and control stage | CW-MC |
| Demolition waste impacts | WI |
| Demolition waste impacts on environment | WI-EN |
| Demolition waste impacts on society | WI-SO |
| Demolition waste impacts on economy | WI-EC |
| Demolition waste challenges | WC |
| Demolition waste challenges for waste handling | WC-WH |
| Demolition waste challenges for awareness and training | WC-AT |
| Demolition waste challenges for policy and legislation | WC-PL |

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
