# Peer review of "Environmental Management Framework for Road Network Demolition Wastes for Construction Industry of Pakistan"

_sustainability, doi:10.3390/su16104302_

Round 1
Reviewer 1 Report
Comments and Suggestions for Authors
This is a very interesting manuscript. Please make the following revisions to promote its publication.
(1) In the garbage generated by the construction industry, it seems that the garbage from road engineering is not the main content. In addition, at present, a large number of construction waste or tailings are used in subgrade engineering.
(2) Whether "Pakistan" can be added to the title of the manuscript. Because the questionnaire survey of the manuscript seems to be aimed at experts or engineers from Pakistan. At present, it seems that the research results of the manuscript are difficult to be extended to other countries or cities around the world. As far as I can see, Islamabad belongs to Pakistan. Please forgive me if my superficial knowledge of geography leads to mistakes.
(3) Literature review of PLS-SEM is added in the Section 2. Because it is the main research method of this manuscript.
(4) In the Section 3, there are too many descriptions of statistical common sense in the "Delphi method". Suggest deletion. In addition, the main visiting procedure or visiting outline of Delphi method should be added. This is where readers are more interested.
(5) The theoretical basis or documentary evidence of the three hypotheses needs to be disclosed more.
(6) Based on the analysis results of PLS-SEM, can you give some strategies to improve the environmental management of road network? The current proposal is too weak.
(7) Can you further describe the difference between construction waste management and road engineering waste management? This will help readers to better understand the contribution of this manuscript.
Comments on the Quality of English LanguageMinor editing of English language required.
Author Response
Comment:
This is a very interesting manuscript. Please make the following revisions to promote its publication.
Compliance:
Authors are very thankful for your valuable comments which has helped and guided to improve the work. As advised, the comments are addressed as bellow
Comment No. 01:
In the garbage generated by the construction industry, it seems that the garbage from road engineering is not the main content. In addition, at present, a large number of construction waste or tailings are used in subgrade engineering.
Compliance:
The concern of respected reviewer is agreed as the road demolition waste is not only the waste generated by construction sector. Since the current the work limits itself to the road networks, that is why on this portion of construction waste is targeted. Since the research current work is in progress, the valuable comments are noted to be incorporated when the later phases of life cycle will be investigated.
Comment No. 02:
Whether "Pakistan" can be added to the title of the manuscript. Because the questionnaire survey of the manuscript seems to be aimed at experts or engineers from Pakistan. At present, it seems that the research results of the manuscript are difficult to be extended to other countries or cities around the world. As far as I can see, Islamabad belongs to Pakistan. Please forgive me if my superficial knowledge of geography leads to mistakes.
Compliance:
The concern of respected reviewer is agreed and the word “Pakistan” has been added in the title.
Comment No. 03:
Literature review of PLS-SEM is added in the Section 2. Because it is the main research method of this manuscript.
Compliance:
As advised, the literature review has been added accordingly.
Comment No. 04:
In the Section 3, there are too many descriptions of statistical common sense in the "Delphi method". Suggest deletion. In addition, the main visiting procedure or visiting outline of Delphi method should be added. This is where readers are more interested.
Compliance:
Authors are thankful for valuable comment. As suggested, certain description have been deleted. The outline of Delphi technique has been incorporated in figure-1, Page-11 to interest the readers.
Comment No. 05:
The theoretical basis or documentary evidence of the three hypotheses needs to be disclosed more.
Compliance:
As advised, the theoretical basis or documentary evidence of the three hypotheses are added in section 2, page 22&23 as below :
Hypothesis 1: There is a significant impact of causes of demolition waste generation on Demolition waste impacts.
Theoretical Basis: This hypothesis is grounded in the understanding that the causes of demolition waste generation, such as construction practices, material choices, and project management decisions, can have significant implications for the environmental, social, and economic impacts of demolition activities. For example, inefficient construction practices, poor waste segregation, and lack of recycling infrastructure can lead to higher waste generation rates and increased environmental degradation. Theoretical frameworks such as life cycle assessment (LCA) and environmental impact assessment (EIA) provide a basis for understanding the relationships between demolition waste generation causes and their impacts.
Documentary Evidence: Research studies utilizing LCA, EIA, and other impact assessment methodologies have documented the environmental, social, and economic impacts of demolition waste generation. These studies often analyze the environmental footprint of demolition activities, including greenhouse gas emissions, energy consumption, air and water pollution, and habitat destruction. Additionally, case studies and industry reports provide evidence of specific causes of demolition waste generation, such as building design, construction methods, material selection, and project management practices, and their associated impacts on waste generation and disposal.
Hypothesis 2: There is a significant impact of causes of demolition waste generation on Demolition waste management challenges.
Theoretical Basis: This hypothesis is based on the recognition that the causes of demolition waste generation can create significant challenges for waste management practices and infrastructure. Factors such as the heterogeneity of demolition waste streams, contamination levels, and regulatory requirements can complicate waste sorting, recycling, and disposal processes. Theoretical perspectives from waste management and environmental governance highlight the importance of addressing upstream factors that influence waste generation to effectively manage waste downstream.
Documentary Evidence: Research studies and industry reports have identified various challenges associated with the causes of demolition waste generation. These challenges include inadequate waste characterization and sorting facilities, limited recycling capacity, lack of market demand for recycled materials, and regulatory barriers to waste diversion and recycling. Case studies and best practice guides provide evidence of successful strategies for overcoming these challenges, such as improved waste sorting techniques, investment in recycling infrastructure, and policy incentives for sustainable demolition practices.
Hypothesis 3: There is a significant impact of demolition waste impacts on Demolition waste management challenges.
Theoretical Basis: This hypothesis posits that the environmental, social, and economic impacts of demolition waste, such as pollution, habitat destruction, public health risks, and economic costs, can exacerbate existing challenges in waste management practices. Theoretical perspectives from environmental sociology, risk assessment, and policy studies emphasize the interconnectedness of environmental problems and the need for integrated approaches to address complex challenges.
Documentary Evidence: Research studies and policy reports have documented the impacts of demolition waste on waste management challenges. These impacts may include increased landfill pressure, regulatory compliance costs, public opposition to waste facilities, and health and safety risks for waste workers and surrounding communities. Case studies and empirical research provide evidence of the linkages between demolition waste impacts and waste management challenges, highlighting the need for holistic and proactive approaches to waste management and environmental protection.
Comment No. 06:
Based on the analysis results of PLS-SEM, can you give some strategies to improve the environmental management of road network? The current proposal is too weak.
Compliance:
Authors are thankful for the comments. Based upon the results of PLS-SEM to understand factors influencing the environmental management of road demolition waste, the following strategies are proposed and included in section-4, page 32.
Comment No. 07:
Can you further describe the difference between construction waste management and road engineering waste management? This will help readers to better understand the contribution of this manuscript.
Compliance:
The valuable comment is acknowledged. As advised a comprehensive difference is provided in table-1,section-1,page 3.
Reviewer 2 Report
Comments and Suggestions for Authors
1- The novelty of the study should be explained.
2- The benefits of the study from economic views should be explained.
3- How the developed system can be applied for the other cases. Is this a comprehensive method in pollution minimization.
Comments on the Quality of English Language
The English language should be modified.
Author Response
Authors are very thankful for valuable comments which has helped and guided to improve the work. As advised, the comments are addressed as below:
Comment No. 01:
The novelty of the study should be explained.
Compliance:
The current work presents the novelty in the following terms:
- Based upon the critical literature review, there exists a research gap for environmental management specific to road demolition waste. This becomes more concerned in case of developing countries having rapid urbanization and population growth where road network is key to the social and as well economic uplift.
- A comprehensive management for all phases of life cycle also seems missing.
- A numerical based correlation between various variables responsible for environmental management for infrastructure facilities, specifically road network, is also missing to support the sustainable demolition waste management practices.
Comment No. 02:
The benefits of the study from economic views should be explained.
Compliance:
The economic benefits of managing demolition waste from roads stem from cost savings, revenue generation, resource conservation, compliance with regulations, and the promotion of corporate social responsibility. By adopting sustainable waste management practices, construction companies can optimize economic efficiency, reduce environmental impact, and contribute to the long-term viability of infrastructure projects. The following benefits form economic point of view have been incorporated in section-3, Page-31.
- Proper management of road demolition waste can help reduce disposal costs associated with landfilling or incineration, thus avoiding disposal expenditures.
- Effective management of road demolition waste ensures compliance with environmental regulations and standards, avoiding potential fines or penalties for non-compliance. By adhering to waste management regulations, construction companies can avoid costly legal disputes and reputational damage.
- Implementing sustainable waste management practices for road demolition waste demonstrates a commitment to corporate social responsibility (CSR). This can enhance the reputation of construction companies, attract environmentally conscious clients, and create opportunities for partnerships and collaborations. Thus, supporting a healthy and sustainable economic competition.
- This study will guide to conserves valuable resources and minimizes the environmental impact of resource extraction, contributing to long-term economic sustainability.
Comment No. 03:
How the developed system can be applied for the other cases. Is this a comprehensive method in pollution minimization.
Compliance:
The concern of respected reviewer is appreciated. Currently, the study is in its initial phase and valuable suggestion has been noted. As per authors understanding, the developed system for managing demolition waste from roads can serve as a model or framework that can be support other cases within the construction industry or beyond with some modifications according to their circumstances. However, the proposed framework may not be able to address all aspects of pollution minimization comprehensively on its own. It can only provide a solid foundation for promoting sustainable practices and reducing environmental impact. By applying the principles of waste reduction, recycling, pollution prevention, and continuous improvement, the system can contribute to broader efforts to minimize pollution and promote environmental sustainability across various sectors and contexts.
Reviewer 3 Report
Comments and Suggestions for Authors
Dear author, please see the comments in the attached file.

Author Response
Authors are very thankful for valuable comments which has helped and guided to improve the work. As advised, the comments are addressed as below:
Comment No. 01:
It is not appropriate to put acronym in title (PLS-SEM).
Compliance:
As advised, the acronym in title (PLS-SEM) is removed and title has been modified
Comment No. 02:
In the abstract, it is not recommended to use acronyms such as "DWM" and "SEM", It is better to simplify the sentence so that it will be easier to understand.
Compliance:
As advised, the acronym such as "DWM" and "SEM" in title (PLS-SEM) have been removed from the abstract.
Comment No. 03:
Keywords are keywords, not sentences...do not use more than two words.
Compliance:
The keywords have been revised as advised.
Comment No. 04:
Please highlight the most important originalities of the research in the end introduction.
Compliance:
The authors are thankful for valuable suggestion. The following important originalities have been incorporated as advised.
“Based upon the critical literature review, there exists a research gap for environmental management specific to road demolition waste. This becomes more concerned in case of developing countries having rapid urbanization and population growth where road network is key to the social and as well economic uplift. A comprehensive management for all phases of life cycle also seems missing. A numerical based correlation between various variables responsible for environmental management for infrastructure facilities, specifically road network, is also missing to support the sustainable demolition waste management practices.
Thus, the current work aims to support a robust framework by encompassing critical analysis innovative methodologies, cross-disciplinary insights and empirical validation, critical analysis to promote a sustainable change for serious issue of waste management in road construction industry”.
Comment No. 05:
I found the research methodology very well explained, the variable measures were carefully selected, they mentioned the experts involved and the method used to achieve the consensus of the experts. Experts with more than 20 years of experience were also involved in the pilot study, helping to design the conceptual framework for the research study. Therefore and as mentioned in conclusions, the measurement tools are reliable. In the results section, I suggest adding the graphic legends and the titles of the graphs for FIG 6- Demographic classification, mentioned in lines 482 and 483.
Compliance:
As advised by the respected reviewer, the legend and tittles for Fig 6 has been updated.
Comment No. 06:
Line 289, page. 8 "research employed an action of research" this expression is redundant.
Compliance:
As advised by the respected reviewer, the redundant line has been amended accordingly.
Comment No. 07:
Conclusions are coherent with the objective mentioned in the introduction, It is mentioned that all the relationships evaluated were considered significant, thus supporting all the hypotheses raised in the study, Summary, it was possible to explore and confirm the relationship between the causes, impacts and challenges of demolition waste management, as proposed in the objective of the study.
Compliance:
The authors are thankful for valuable suggestion. In order to further clarify about the interrelationship between the variables, the following explanations have been added in section-4, Page-32.
Reviewer 4 Report
Comments and Suggestions for Authors
The present manuscript is very interesting because it generates a framework for a better understanding of the sustainable management of demolition waste. However, the following aspects should be considered to complement the manuscript:
1. In the methodological section, the statistical techniques used in the entire work must be established. Various statistical tests are mentioned in the results, but they were not described in the methodology.
2. In the results section, knowledge of certain statistical techniques is assumed and values, cut-off thresholds, hypothesis tests, VIF, Cronbach's alpha, etc. are mentioned. Strictly speaking, these values should be indicated in the methodological section.
3. The study area is briefly mentioned. Therefore, it is not clear how useful the results obtained will be in the study area and elsewhere.
4. It is suggested to separate the “Conclusions and Recommendations” section. The recommendations section should include information that addresses the practical applications that this work will have.
5. Section 3.2 Partial Least Squares Structural Equation Modeling (PLS-SEM) should be more robust and explained in mathematical and statistical terms.
Author Response
Comment:
The present manuscript is very interesting because it generates a framework for a better understanding of the sustainable management of demolition waste. However, the following aspects should be considered to complement the manuscript:
Compliance:
Authors are very thankful for valuable comments which has helped and guided to improve the work. As advised, the comments are addressed as below:
Comment No. 01:
In the methodological section, the statistical techniques used in the entire work must be established. Various statistical tests are mentioned in the results, but they were not described in the methodology.
Compliance:
As advised, various statistical tests adopted in the study have been described in section-2, Page-20, Table-8 is also added to strengthen the clarifications.
Comment No. 02:
In the results section, knowledge of certain statistical techniques is assumed and values, cut-off thresholds, hypothesis tests, VIF, Cronbach's alpha, etc. are mentioned. Strictly speaking, these values should be indicated in the methodological section.
Compliance:
As advised, the values of statistical techniques have been provided in methodology section-2, Page-20, Table-8
Comment No. 03:
The study area is briefly mentioned. Therefore, it is not clear how useful the results obtained will be in the study area and elsewhere.
Compliance:
The authors are thankful for valuable suggestion. The following usefulness of results has been added in section-3,Page-30
“The economic benefits of managing demolition waste from roads stem from cost savings, revenue generation, resource conservation, compliance with regulations, and the promotion of corporate social responsibility. By adopting sustainable waste management practices, construction companies can optimize economic efficiency, reduce environmental impact, and contribute to the long-term viability of infrastructure projects”
Comment No. 04:
It is suggested to separate the “Conclusions and Recommendations” section. The recommendations section should include information that addresses the practical applications that this work will have.
Compliance:
The authors are thankful for valuable suggestion. The “Conclusions and Recommendations” section have been separated. In addition, the following practical applications for the research work have been added in section-4, Page -31.
- Proper management of road demolition waste can help reduce disposal costs associated with landfilling or incineration, thus avoiding disposal expenditures.
- Effective management of road demolition waste ensures compliance with environmental regulations and standards, avoiding potential fines or penalties for non-compliance. By adhering to waste management regulations, construction companies can avoid costly legal disputes and reputational damage.
- Implementing sustainable waste management practices for road demolition waste demonstrates a commitment to corporate social responsibility (CSR). This can enhance the reputation of construction companies, attract environmentally conscious clients, and create opportunities for partnerships and collaborations. Thus, supporting a healthy and sustainable economic competition.
- This study will guide to conserves valuable resources and minimizes the environmental impact of resource extraction, contributing to long-term economic sustainability.
Comment No. 05:
Section 3.2 Partial Least Squares Structural Equation Modeling (PLS-SEM) should be more robust and explained in mathematical and statistical terms.
Compliance:
As advised, a comprehensive elaboration has been provided in tabular form, section-2,Page20, Table-8. The ranges have also been provide in the same section-2,Page 20, Table-8.
Reviewer 5 Report
Comments and Suggestions for Authors
Dear Author's, below I sent comments, which could improve your paper:
1) Chapters 1 and 2 should be merge into 1 called Introduction.
2) The litearture review is very long. Please correct them in case of research purpose. Also the research purpose isn't clearly described. What about literature review connected with waste modelling?
3) Coding gudielins should be presented as Appendix.
4) On fig. 2 I suggest presentation of examples codes from hierarchy. Presented plot is very hard to read.
5) In the paper I couldn't find information why You choose PLS-SEm method.
6) Please describe How You connect proposed code categoreis with PLS-SEM. How You provide validation of results.
7) Conclusion and recommendation section should be rewritten. You present conclusions, which were presented in chapter 4. Please add information about usefulness of presented model, what limitations it provides, how big data collection is needed for properly calculation etc.
Author Response
Authors are very thankful for valuable comments which has helped and guided to improve the work. As advised, the comments are addressed as below:
Comment No. 01:
Chapters 1 and 2 should be merge into 1 called Introduction.
Compliance:
As advised, section 1 and 2 have been merged into a single section “Introduction” .
Comment No. 02:
The literature review is very long. Please correct them in case of research purpose. Also the research purpose isn't clearly described. What about literature review connected with waste modelling?
Compliance:
As advised, the literature is shortened and aligned with the research purpose which has been added in section -1, page 9&10 as follows:
“Based upon the critical literature review, there exists a research gap for environmental management specific to road demolition waste. This becomes more concerned in case of developing countries having rapid urbanization and population growth where road network is key to the social and as well economic uplift. A comprehensive management for all phases of life cycle also seems missing. A numerical based correlation between various variables responsible for environmental management for infrastructure facilities, specifically road network, is also missing to support the sustainable demolition waste management practices.
Thus, the current work aims to support a robust framework by encompassing critical analysis innovative methodologies, cross-disciplinary insights and empirical validation, critical analysis to promote a sustainable change for serious issue of waste management in road construction industry”.
Comment No. 03:
Coding guidelines should be presented as Appendix.
Compliance:
As advised, Appendix A has been added and the coding guide lines have presented accordingly.
Comment No. 04:
On fig. 2, I suggest presentation of examples codes from hierarchy. Presented plot is very hard to read.
Compliance:
The concern of respected reviewer is acknowledged. The then number Fig-2 is changed to Fig 3 has been improved to address the advised concern. Please refer page-17.
Comment No. 05:
In the paper I couldn't find information why You choose PLS-SEM method.
Compliance:
The concern of respected reviewer is agreed. The following justification has been added in section- 2, Page- 18.
“ Choosing Partial Least Squares Structural Equation Modeling (PLS-SEM) as the methodological framework for studying demolition waste management is motivated by several factors. Firstly, the complexity of variables involved in waste management, such as waste generation rates and recycling efficiency, can be effectively handled by PLS-SEM, which captures relationships between latent constructs and observed variables. Secondly, PLS-SEM's robustness with limited sample sizes makes it suitable for studies focusing on specific regions or timeframes where data may be scarce. Thirdly, the method's flexibility with non-normally distributed data aligns with the diverse sources often encountered in waste management research. Additionally, PLS-SEM's ability to model formative constructs is advantageous for capturing variables like "waste management practices" or "sustainability initiatives." Its predictive modeling capability supports forecasting waste management outcomes and evaluating strategy effectiveness. Moreover, PLS-SEM can integrate stakeholder perspectives and allows iterative model refinement, crucial in dynamic environments like waste management. Lastly, its clear and interpretable results enhance accessibility for stakeholders involved in decision-making. Overall, employing PLS-SEM facilitates a comprehensive understanding of demolition waste management dynamics, fostering sustainable and efficient practices [1].
Comment No. 06:
Please describe How You connect proposed code categories with PLS-SEM. How You provide validation of results.
Compliance:
The authors are thankful for the comment. It is submitted that the connection code categories is presented in Research model, figure -4, Page-18. The validation has also been achieved through the various statistical techniques as elaborated in section “Methodology, table-8”. However, the following further elaboration has been added in section “methodology, page-19”
“Connecting proposed code categories with Partial Least Squares Structural Equation Modeling (PLS-SEM) involves integrating qualitative data analysis with quantitative modeling. This integration begins by treating each code category developed through qualitative analysis as a latent variable in the PLS-SEM structural model. These variables represent constructs or dimensions relevant to the research question. The relationships between categories and other variables in the model are then specified, guided by theoretical grounding and informed by the qualitative analysis. Quantitative measures derived from the coded data serve as indicators for these latent variables, providing a quantitative representation of the underlying constructs [2].”
Comment No. 07:
Conclusion and recommendation section should be rewritten. You present conclusions, which were presented in chapter 4. Please add information about usefulness of presented model, what limitations it provides, how big data collection is needed for properly calculation etc.
Compliance:
As advised, necessary modification have been made in section “Conclusion and recommendation”. The useful of the model has been elaborated in section-1, page 9&10. Limitation have been addressed in section-4, Page-31 as follows
“ As per authors understanding, the developed system for managing demolition waste from roads can serve as a model or framework that can be support other cases within the construction industry or beyond with some modifications according to their circumstances. However, the proposed framework may not be able to address all aspects of pollution minimization comprehensively on its own. It can only provide a solid foundation for promoting sustainable practices and reducing environmental impact. By applying the principles of waste reduction, recycling, pollution prevention, and continuous improvement, the system can contribute to broader efforts to minimize pollution and promote environmental sustainability across various sectors and contexts” .
- Cepeda-Carrion, G., J.-G. Cegarra-Navarro, and V. Cillo, Tips to use partial least squares structural equation modelling (PLS-SEM) in knowledge management. Journal of Knowledge Management, 2019. 23(1): p. 67-89.
- Marsh, H.W. and K.-T. Hau, Applications of latent-variable models in educational psychology: The need for methodological-substantive synergies. Contemporary educational psychology, 2007. 32(1): p. 151-170.
Reviewer 6 Report
Comments and Suggestions for Authors
The article builds an environmental management framework for road network demolition wastes based on the PLS-SEM methodology, and a lot of work has been done. However, reviewer believes that there are still many issues with the writing of key content and the standardization of writing in the paper. The authors are suggested to further revise the manuscript. Some suggestions:
1. The boundaries between the various parts of the article's abstract are blurred, resulting in a less logical abstract. And the authors mixed the research work and conclusions together, which also makes the hierarchy not clear. It is recommended that the authors first state the overall research work, and then state the research conclusions one by one based on the research content.
2. What is the purpose of setting up a separate literature review? Generally, it is included in the section of Introduction, which summarizes the main development status, extracting problems, and then elaborates on the problems that this study aims to solve. And the literature review is relatively scattered.
3. The research work carried out in this article needs to be described in detail, and the research methods should correspond intuitively with the research content.
4. The numbers ”7”, ”8”, ”9” and ”11” in figure 1 need to be explained in the text. The layout and formatting of the whole figure could also be optimized.
5. Figure 6 lacks a textual counterpart in the text. There could be a lot of things that could be analyzed in Figure 6, but the author simply ignores this part. In addition, the names of the horizontal and vertical coordinates are missing in Figure 6.
6. Questionnaires were an important method in this study, but only 120 individuals were surveyed, yielding 94 pieces of usable data. This sample size seems a little too small.
7. Most of the images in the text need further adjustment, e.g., Figure 2 and Figure 6 are drawn with low resolution and not clear. Figures 3 and 7 are not very aesthetically pleasing, and Figure 7 even obscures some of the text.
8. Please concise the conclusions, highlight important findings, and the expression of the conclusion needs to be logical.
Author Response
Comment:
The article builds an environmental management framework for road network demolition wastes based on the PLS-SEM methodology, and a lot of work has been done. However, reviewer believes that there are still many issues with the writing of key content and the standardization of writing in the paper. The authors are suggested to further revise the manuscript.
Compliance:
Authors are very thankful for valuable comments which has helped and guided to improve the work. As advised, the comments are addressed as below:
Comment No. 01:
The boundaries between the various parts of the article's abstract are blurred, resulting in a less logical abstract. And the authors mixed the research work and conclusions together, which also makes the hierarchy not clear. It is recommended that the authors first state the overall research work, and then state the research conclusions one by one based on the research content.
Compliance:
As advised, various section of the work have been modified and rearranged.
Comment No. 02:
What is the purpose of setting up a separate literature review? Generally, it is included in the section of Introduction, which summarizes the main development status, extracting problems, and then elaborates on the problems that this study aims to solve. And the literature review is relatively scattered.
Compliance:
In compliance with the respected viewer, the section 1 Introduction and section 2 Literature Review have been merged and modified.
Comment No. 03:
The research work carried out in this article needs to be described in detail, and the research methods should correspond intuitively with the research content.
Compliance:
The research Methodology section has been critically reviewed and modified.
Comment No. 04:
The numbers ”7”, ”8”, ”9” and ”11” in figure 1 need to be explained in the text. The layout and formatting of the whole figure could also be optimized.
Compliance:
The comment of respected reviewer is acknowledged. The numbers ”7”, ”8”, ”9” and ”11” in figure 1 has been elaborated in section-2 ,Page-13. Figures have also been modified.
Comment No. 05:
Figure 6 lacks a textual counterpart in the text. There could be a lot of things that could be analyzed in Figure 6, but the author simply ignores this part. In addition, the names of the horizontal and vertical coordinates are missing in Figure 6.
Compliance:
As advised, necessary modification have been made in Fig-6 (New Fig-9) in form of text, Horizontal and vertical coordinates have been added and also explained in detail in Section-3, Page-24.
Comment No. 06:
Questionnaires were an important method in this study, but only 120 individuals were surveyed, yielding 94 pieces of usable data. This sample size seems a little too small.
Compliance:
The concern of respected reviewer is noted. It is submitted that sample size was supported by previous studies where same technique was adopted by previous researchers.
Comment No. 07:
Most of the images in the text need further adjustment, e.g., Figure 2 and Figure 6 are drawn with low resolution and not clear. Figures 3 and 7 are not very aesthetically pleasing, and Figure 7 even obscures some of the text.
Compliance:
Authors are thankful for the valuable comment. As advised, the figures have been corrected.
Comment No. 08:
Please concise the conclusions, highlight important findings, and the expression of the conclusion needs to be logical.
Compliance:
As advised, necessary compliance has been made.
Round 2
Reviewer 1 Report
Comments and Suggestions for Authors
Almost all my suggestions and questions have been well explained. The unremitting efforts of the authors prompted me to recommend the publication of this manuscript. Congratulations to the author.
Reviewer 2 Report
Comments and Suggestions for Authors
The paper can be accepted.
Reviewer 4 Report
Comments and Suggestions for Authors
The authors addressed all comments made to the manuscript. New sections, new tables and information were added that allow this manuscript to be very useful for the public interested in the topic of construction and demolition waste. I have no further comments to make.
Reviewer 6 Report
Comments and Suggestions for Authors
The manuscript has been well revised, and it is recommended to be accepted for publication.